# Viewpoint Planning for Range Sensors Using Feature Cluster Constrained Spaces for Robot Vision Systems

**DOI:** 10.3390/s23187964

**Published:** 2023-09-18

**Authors:** Alejandro Magaña, Michiel Vlaeyen, Han Haitjema, Philipp Bauer, Benedikt Schmucker, Gunther Reinhart

**Affiliations:** 1Institute for Machine Tools and Industrial Management, Technical University of Munich, 85747 Garching, Germany; 2Department of Mechanical Engineering, KU Leuven, 3001 Leuven, Belgium; 3Flanders Make—Core Lab MaPS, KU Leuven, 3001 Leuven, Belgium

**Keywords:** viewpoint planning problem, vision system automation, constraint planning, robot vision system, range sensor

## Abstract

The efficient computation of viewpoints for solving vision tasks comprising multi-features (regions of interest) represents a common challenge that any robot vision system (RVS) using range sensors faces. The characterization of valid and robust viewpoints is even more complex within real applications that require the consideration of various system constraints and model uncertainties. Hence, to address some of the challenges, our previous work outlined the computation of valid viewpoints as a geometrical problem and proposed feature-based constrained spaces (C-spaces) to tackle this problem efficiently for acquiring one feature. The present paper extends the concept of C-spaces to consider multi-feature problems using feature cluster constrained spaces (${}^{G}C$-spaces). A ${}^{G}C$-space represents a closed-form, geometrical solution that provides an infinite set of valid viewpoints for acquiring a cluster of features satisfying diverse viewpoint constraints. Furthermore, the current study outlines a generic viewpoint planning strategy based on ${}^{G}C$-spaces for solving vision tasks comprising multi-feature scenarios effectively and efficiently. The applicability of the proposed framework is validated on two different industrial vision systems used for dimensional metrology tasks.

## 1. Introduction

The number of applications requiring machine vision capabilities (e.g., image-based inspection, object digitization, scene exploration, object detection, visual servoing, robot calibration, mobile navigation) has rapidly increased within the research and industry over the last decade [1,2]. Depending on the vision task and system complexity, the automation of such applications is necessary to provide effective and efficient solutions [3]. Over the past decade, robot vision systems (RVS) equipped with a range sensor have proven useful for automating these tasks. However, programming RVS to perform such tasks is considered a tedious and complex challenge. Programmers face a common challenge: the computation of necessary and valid viewpoints to perform the required vision task. This challenge is well known as the view(point) planning problem (VPP).

In an attempt to propose an efficient and generic framework to tackle the VPP, this publication extends the concept of feature-based constrained spaces (C-spaces), introduced in our previous publication [4], and outlines a viewpoint planning strategy for multi-feature scenarios using feature cluster constrained spaces (CG-spaces).

### 1.1. ${}^{G}C$-spaces for Solving the Viewpoint Planning Problem

The VPP can be comprehended by considering the following simple and generic formulation (F):

**Problem** **1.**
*What is the minimum number of viewpoints to acquire a set of features considering different viewpoint constraints?*


Figure 1 provides a graphical representation of this problem for acquiring a set of four features. Solving the VPP based on this formulation would require a monolithic strategy to efficiently solve a high-dimensional problem where multiple viewpoint constraints must be satisfied simultaneously. As a first step, we believe that a suitable reformulation and modularization of the VPP is necessary to reduce its complexity and outline more efficient solutions. In the present study, we propose first breaking the VPP into two subproblems, i.e., the viewpoint generation problem (VGP) and the set cover problem (SCP). The detailed reformulation of the VPP and its subproblems is handled within Section 3.

Magaña et al. [4] focused on the first subproblem of the VPP, i.e., the VGP, which addresses the generation of valid viewpoints to acquire a single feature. The VGP was posed as a pure geometrical challenge that can be solved effectively using feature-based constrained viewpoint spaces (C-spaces). A C-space represents the spatial solution space in 6D (translation and rotation) that a set of viewpoint constraints spans. Therefore, C-spaces represent an analytical, geometric, and closed-form solution that provides infinite valid viewpoints to capture a feature. Especially within real applications, infinite solution spaces are advantageous for choosing seamlessly alternative solutions to compensate for model uncertainties or unknown constraints affecting the validity of a chosen viewpoint.

Although [4] delivered a sound solution for computing valid viewpoints and demonstrated the potential of C-spaces for a simplified multi-feature acquisition, the authors did not pose the VGP as a multiple-feature problem. Moreover, a strategy for finding potential features that could be acquired together was not proposed. For example, it is generally not obvious which features could be acquired together in more complex scenarios with several features, such as the one depicted in Figure 1.

For these reasons, the present paper first extends the concept of C-spaces for acquiring multiple features by introducing CG-spaces. CG-spaces can be analogously interpreted as spatial, continuous solution spaces spanned by a set of C-spaces for acquiring a group of features. Hence, any sensor pose within it fulfills all viewpoint constraints $\tilde{C}$ and can be considered valid for capturing all regarded features. Figure 1 provides an exemplary and simplified illustration of the VPP and the spanned CG-spaces for acquiring four features. The VPP can be easily solved by selecting one sensor pose from each CG-space. Section 4.1 outlines the characterization of CG-spaces based on C-spaces.

In a second step, the present work addresses the second subproblem of the VPP, i.e., the SCP, and introduces a viewpoint plan strategy in Section 4.2 to identify the potential features that can be clustered together. Having identified the minimum number of necessary feature clusters, hence, viewpoints, the corresponding CG-spaces can then be computed to solve these multi-viewpoint tasks.

In the last Section 5, the applicability and potential of the proposed approach are evaluated in the context of dimensional metrology tasks using two industrial vision systems with different range sensors.

### 1.2. Related Work

The VPP has been the focus of the research over the last three decades within a wide range of vision tasks that demand the computation of generalized viewpoints. A broad overview of the overall progress, challenges, and applications of the VPP is provided within various surveys [2,5,6,7,8,9]. Depending on the a priori knowledge required, the approaches for viewpoint planning can be roughly classified into *model-based* and *non-model-based* approaches. Since our framework falls under the first category, this section provides a more comprehensive overview of the related research following model-based approaches.

Model-based methods require minimal spatial information from the features, e.g., its position and orientation, can be categorized as *synthesis* or *sampling-based* approaches. On the one hand, synthesis approaches consider first the characterization of a continuous or discrete solution space. On the other hand, sampling-based techniques generate and evaluate viewpoints based on objective functions without the need to span a search space.

#### 1.2.1. Synthesis

The synthesis of solution spaces (related terms: C-spaces and CG-spaces, search space, configuration space, viewpoint space, visibility map, visibility matrix, visibility volumes, imaging space, scannability frustum, visual hull) was posed in the first studies addressing the VPP and has the advantage of providing a straightforward comprehension and spatial interpretation of the general problem.

The pioneering studies of [10,11] are considered among the first to consider the characterization of a continuous viewing space in $R^{3}$. Their works proposed analytical relationships to characterize multiple constraints geometrically for 2D sensors. However, their research concentrated on generating single viewpoints and did not consider a strategy for more complex applications requiring multiple viewpoints. Based on the analytical findings provided by the previous work of [10,11], Tarabanis et al. [5,12] introduced a model-based sensor planning system that characterizes a continuous search space taking into account the occlusion constraints. The rest of the constraints are formalized based on objective functions.

The study of [13] extended and addressed some of the drawbacks of [5], such as the non-linearity and convergence guarantee of the objective functions. In their work, they opted to synthesize the 3D search spaces, as suggested by [10]. Since not all viewpoint constraints could be synthesized, the authors proposed a gradient-based approach to find valid sensor poses within the search space and to evaluate the rest of the constraints. In parallel, in a series of publications, Reed et al. [14,15] also followed an explicit characterization of the viewpoint space in $R^{3}$ for range sensors integrating imaging, occlusion, and workspace constraints. However, the final selection of valid viewpoints required a discretization of the search space for finding a proper sensor orientation.

Another significant line of research was introduced by [16], who proposed characterizing a discretized viewpoint space. Analogously, ref. [17] introduced the measurability matrix, which extended the visibility matrix of [16] to three dimensions under the consideration of further constraints. Both publications [18,19] placed the final selection of viewpoints in the context of the SCP and considered various heuristic algorithms for finding optimal viewpoints. More recent works [20,21,22] followed and extended the work of [17], confirming the usability of visibility matrices and modularization of the VPP.

Furthermore, a handful of works [23,24,25,26,27,28,29] also considered the explicit characterization of search spaces for some specific constraints. Similar to others, the search for viable viewpoint candidates was performed using different optimization algorithms to assess the satisfiability of other constraints. Some of these studies, e.g., [24,27,28], also suggested identifying potential groups of features or surfaces first based on their positions and orientations before spanning a search space to increase the planning efficiency.

#### 1.2.2. Sampling-Based

Unlike synthesis methods, sampling-based approaches do not rely on the explicit characterization of a search space. They evaluate the validity of each candidate viewpoint based on objective functions for each constraint on the viewpoint, which are solved by metaheuristic optimization algorithms, e.g., simulated annealing or evolutionary algorithms [30,31,32,33,34]. These approaches focus on the efficient formulation of objective functions to satisfy the viewpoint constraints and find valid solutions. By neglecting the characterization of a search space, which can be computationally expensive, sampling methods can be especially efficient within simple scenarios considering only a few constraints and features.

### 1.3. Need for Action

Due to the heterogeneity of vision tasks and RVS confronted with the VPP, many researchers (see Section 1.2) have presented tailor-made and sound solutions to address this problem. To our knowledge, a standard approach for solving the VPP has not yet been established within the research or the industry.

Our literature review revealed various ways of characterizing solution spaces to identify valid viewpoints that satisfy the VPP, either explicitly or implicitly. Implicit methods incorporating sampling techniques can be highly advantageous and computationally efficient, especially in straightforward scenarios with fewer constraints. However, modeling uncertainties to compensate for robust solutions in complex applications that consider multiple features and constraints poses a more challenging task when using objective functions. That is mainly because the validation of objective functions must be explicitly proofed for each computed viewpoint, and no alternative viewpoints are initially proposed. This point is particularly crucial in real-world applications, where modeling uncertainties cannot be ignored, and robust approaches for compensating for these uncertainties are necessary. The difficulties of handling the VPP by discretizing the valid viewpoint space can be mitigated by using analytical modeling to establish the validity of constraints in special Euclidean explicitly. Therefore, an approach that proposes modeling every constraint as a solution space has the potential to address the VPP explicitly. However, addressing the problem this way requires an extensive analysis of the constraints and how they can be spatially modeled. While our previous work [4] addresses this challenge for single features, the current study extends the concept to multiple features and outlines a suitable strategy to tackle the VPP holistically considering the following points:Multi-stage formulation: The complexity of the VPP can be subdivided into multiple subproblems. Simple and more efficient solutions can then be individually formulated for each subproblem.Model-based solution: Assuming that a priori information about viewpoint constraints (vision system, object, and task) is available, this knowledge should be used in the most effective and analytical manner. Furthermore, in this study, it is assumed that each viewpoint constraint should be spatially modeled in the special Euclidean (6D) aligned to a synthesis approach (see Section 1.2). This can be carried out by characterizing each constraint as a topological space, i.e., C-space. If all viewpoint constraints can then be modeled as topological spaces and integrated together into CG-spaces, the search for viable candidates can be reduced to the selection of a sensor pose within such spaces.Viewpoint planning strategy: Taking into account the modularization of the VPP and the characterization of CG-spaces, a superordinated, holistic viewpoint planning strategy must be outlined for delivering a final selection of valid viewpoint candidates.

To the best of our knowledge, no other works have utilized a feature-based approach and combined multiple infinite spaces to compute valid viewpoints while considering several constraints. Furthermore, no generic strategies can be employed in conjunction with CG-spaces. For this reason, a tailor-made strategy must be outlined.

### 1.4. Outline

The outline of this research is shown in Figure 2. First, Section 2 introduces a generic domain model for RVSs. Due to the heterogeneity of the definitions of RVSs in different applications, this section limits the scope of the presented research and helps to evaluate the transferability of the proposed models to other systems. Then, the core modules of the framework presented in this report address all these points and are individually addressed in Section 3, Section 4.1 and Section 4.2.

Section 3 revises the formulation of the VPP and its sub-problems. Then, a new generic formulation of the VPP is introduced based on the concept of CG-spaces. Using the introduced formulation of CG-spaces, their characterization is comprehensibly addressed in Section 4.1. Based on the concept of CG-spaces, a holistic viewpoint planning strategy to tackle the VPP is presented in Section 4.2. Based on an academic example, the characterization of CG-spaces and the proposed strategy is verified.

Finally, Section 5 evaluates the validity of CG-spaces and the proposed strategy to solve the VPP using two real RVS and a metrology application. Finally, Section 6 presents a summary and the conclusions of this paper. In addition, a comprehensive data set of multiple Appendix B, e.g.,CG-space meshes, renders, frames, can be found in the attachment of this paper.

## 2. Robot Vision System Domains

This section provides a brief overview of the most elementary components that build an RVS. A more exhaustive description of the models of all components is given in [4]. Figure 3 provides an overview of these elements. The present study considers many variables to describe the domains of a vision system comprehensively. To ease the identification and readability of variables, parameters, vectors, frames, and transformations, the index notation given in Table A2 is used.

### 2.1. Sensor

A sensor (*s*) (synonym: range camera sensor, 3D sensor, imaging system) is defined as a self-contained acquisition device comprising at least two imaging devices $\left\{s_{1},s_{2}\right\}\in \tilde{S}$ (e.g., two cameras or one camera and one coded light source) capable of computing a range image containing depth information. The present study does not explicitly distinguish between the sensing principles of range sensors, e.g., structured light sensors or laser scanners. To ensure the applicability of the outlined solutions with different sensors, a generic and minimal imaging and kinematic model of a generic sensor is presented in this section. Figure A1 illustrates a simplified representation of the sensor’s main components.

#### 2.1.1. Frustum Space

The frustum space (I-space) (related terms: visibility frustum, measurement volume, field-of-view space, sensor workspace) is characterized by a set of various sensor imaging parameters $I_{s}$, such as the horizontal and vertical field of view (FOV) angles $\theta_{s}^{x}$ and $\psi_{s}^{y}$, the middle working distance $d_{s}$, and the near $h_{s}^{near}$ and far $h_{s}^{far}$ viewing planes. The sensor parameters $I_{s}$ describe only the topology of the I-space. The full characterization of the I-space in the special Euclidean requires considering the sensor pose ps given by [4]. In the present study, the pose p of any element is given by its translation t∈R3 and a rotation. The rotation can be given by the *Z-Y-X Euler* angles r=(αz,βy,γx)T or as a rotation matrix R∈R3x3. The pose p∈SE(3) is given in the special Euclidean $SE\left(3\right)=R^{3}\times SO\left(3\right)$, where $SO\left(3\right)\subset R^{3x3}$ denotes the special orthogonal group [35].
(1)Is:=(ps,Is)={ps∈SE(3),(ds,hsnear,hsfar,θsx,ψsy)∈Is}.

The resulting Is manifold spanned by the sensor imaging parameters for a given sensor pose is characterized by its vertices VkIs:=Vk(Is)=(Vkx,Vky,Vkz)T with k=1,…,l and the corresponding joining edges and faces.

#### 2.1.2. Kinematics

The sensor’s kinematic model considers the following frames: $B_{s}^{TCP}$, $B_{s}^{s_{1}}$, and $B_{s}^{s_{2}}$. It is assumed that the frame of the tool center point (TCP) is located at the geometric center of the frustum space and that the frames $B_{s}^{s_{1}}$ and $B_{s}^{s_{2}}$ lie at the reference frame (e.g., the lenses or the focal length) corresponding to the imaging parameters $I_{s}$.

#### 2.1.3. Sensor Orientation

A basic requirement for detecting a surface point is that the relative angle between a sensor and a surface is within the limits of the specific maximum angle of a sensor. The maximal incidence angle is normally provided by the sensor’s manufacturer. This incidence angle, denoted as φfs, is expressed as the angle between the feature’s normal nf and the sensor’s optical axis (*z*-axis) esz and can be calculated as follows:(2)φfsmax>|φfs|,φfs=arccosnf·esz|nf|·|esz|.

Furthermore, the rotation of the sensor around the optical axis is given by the Euler angle $\alpha_{s}^{z}$ (related terms: swing, twist). In many cases, this angle can be arbitrarily chosen. Hence, most of the related works assume a fixed angle during the planning process. However, if the shape of the frustum is asymmetrical, it is reasonable to consider its optimization.

### 2.2. Object

The object (*o*) domain (related terms: object of interest, workpiece, object-, measurement-, inspection-, test-, or probing-object) holds all features to be captured. Since the framework introduced in this publication can be categorized as a feature-based approach, the object may have an arbitrary surface topology. However, if occlusion constraints are taken into account, then a surface model of the object must be considered if the object itself occludes the features.

### 2.3. Features

A feature (*f*) (related terms: region, point or area of interest, inspection feature, key point, entity, object) can be fully specified by its kinematic and geometric parameters, i.e., frame $B_{f}$ and the set of surface points $G_{f}\left(L_{f}\right)$, which depend on a set of geometric lengths $L_{f}$:(3)$$f:=(B_{f},G_{f}\left(L_{f}\right)).$$

Magana et al. [4] proposed the generalization of the feature’s topology using a square with a unique side length $\left\{l_{f}\right\}\in L_{F}$ to describe any 2D feature. This publication also regards this simplification. Moreover, it is assumed that the translation tof and orientation rof of the feature’s origin is given in the object’s coordinate system $B_{o}$ (see Figure 3). Thus, the feature’s frame is given as follows:(4)Bf=Tof(tof,rof).

### 2.4. Robot

In the context of multi-feature scenarios and automation of vision tasks, it is common practice to use a robot (related terms: manipulator, industrial robot, positioning device) for positioning the sensor at the selected viewpoint candidates. Since the viewpoint’s validity may depend on the robot kinematic, [4] considered the robot workspace as a further viewpoint constraint. The present study considers robots to be an optional element of a vision system. Therefore, robot constraints are not further discussed within this publication.

### 2.5. Viewpoint and Viewpoint Constraints

In the literature, there seems to be no general definition of a viewpoint *v*. The present report regards a feature-centered formulation and defines a viewpoint as a triple of the following core elements: a 6D sensor pose ps∈SE(3) to acquire a group of features f∈G considering a set of viewpoint constraints $\tilde{C}$:v:=(G,ps,C˜).

The set of viewpoint constraints $c_{i}\in \tilde{C},i=1,\ldots ,h$ includes all constraints $c_{i}$ affecting the positioning of the sensor. Every constraint $c_{i}$ can be interpreted as a collection of multiple variables of the vision system, e.g., the imaging parameters $I_{s}$, the feature geometry length $l_{f}$, the maximal incidence angle φfsmax. A description of the viewpoint constraints considered in the scope of the current publication is given in Table A1.

### 2.6. Vision Task

A vision task is defined by the set of features *F* (synonyms: feature plan, inspection plan, feature space)
$$f_{m}\in F,m=1,\ldots ,n$$
that must be acquired. A vision task is then fulfilled when there exists a viewpoint plan denoted as *P* that holds a finite number of *k* viewpoints
(Gj,ps,j,C˜)∈P,j=1,…,k,
which guarantees the acquisition of all *n* features from *F* satisfying all viewpoint constraints $\tilde{C}$. Note that $G_{j}$ denotes a subset of features $G_{j}\subseteq F$ and that the union of all subsets corresponds to
$$F=\overset{k}{\underset{j=1}{⋃}}G_{j}.$$

## 3. Formulation of the Viewpoint Planning Problem Based on CG-spaces

We are convinced that a multi-stage formulation of the VPP can reduce its overall complexity and enable more efficient solutions. Therefore, in our first study [4], we proposed to split the VPP into the VGP and SCP and focused on the first one.

Magaña et al. [4] attributes to the VGP (related terms: camera planning, optical camera placement) the calculation of a viewpoint to acquire a single feature considering the fulfillment of a set of viewpoint constraints. Furthermore, when considering a multi-feature application, the efficient selection of multiple viewpoints becomes necessary to accomplish the vision task, which results in a new problem, namely, the SCP. A simplified representation of the VPP and its subproblems are visualized in Figure 4.

### 3.1. The Viewpoint Generation Problem

The generation of viewpoints for acquiring one or multiple features can be treated as an isolated subproblem of the VPP. Hence, our previous study [4] focused on this subproblem, i.e., the VGP, and proposed its consistent and exhaustive formulation. The publication demonstrated that the VGP could be effectively handled as a purely geometrical problem and introduced the concept of C-spaces as the backbone element to generate viewpoints capable of acquiring one feature.

The present study first summarizes the formulation of the VGP in the context of single-feature vision tasks [4]. Then, an extended reformulation of the VGP is placed to consider more complex vision tasks regarding a multi-feature acquisition. In this context, the formal definition of CG-spaces based on C-spaces is introduced as the core element to enable the acquisition of feature clusters.

#### 3.1.1. VGP with C-spaces

The concept of C-spaces can be better understood considering a proper formulation of the VGP [4]:

**Problem** **2.**
*Which is the C-space Cf to acquire a single feature f considering a set of viewpoint constraints $\tilde{C}$?*


The mathematical definition of the C-space denoted as Cf:=C(f,C˜) can now be introduced, considering that there exists a topological space in Cf⊆ SE(3) that holds all valid sensor poses ps to acquire a feature *f*, considering a set of viewpoint constraints $\tilde{C}$. This topological space C is spanned by the Euclidean space $R^{3}$ and the special orthogonal group SO(3) [4]:(5)Cf=R3×SO(3)={ps∈Cf,ps∈SE(3)}={ps(ts,rs)∈Cf∣ts∈R3,rs∈SO(3)}.

Moreover, let the formulation of Cf be further specified and assume that for each viewpoint constraint $\forall c_{i}\in \tilde{C}$ there exists an individual *i* C-space denoted as Cfi:=Ci(f,ci), which represents the topological space where any sensor pose ps∈Cfi satisfies the constraint $c_{i}$. The intersection of all C-spaces conforms to the joint C-space Cf that fulfills all viewpoint constraints:(6)Cf=⋂ci∈C˜Ci(f,ci), and Cf can be considered a subset of any viewpoint constrained space, i.e., Cf⊆Cfi.

An abstract representation of the C-space Cf1 for acquiring $f_{1}$, being constrained by three viewpoint constraints and its corresponding C-spaces Cf11, Cf12, and Cf13, is depicted in Figure 5. Respectively, Cf2 spans the topological space for acquiring feature $f_{2}$, being delimited by two C-spaces, Cf21 and Cf22.

#### 3.1.2. VGP with ${}^{G}C$-spaces

Although the concept of C-spaces provides a sound approach for characterizing an infinite solution space to acquire one feature, [4] did not comprehensively address its scalability in a multi-feature scenario. Hence, this subsection introduces the proper formulation and characterization of CG-spaces to acquire a cluster of features based on C-spaces, considering individual viewpoint constraints for each feature. The concept of C-spaces can then be straightforwardly extended to a multi-feature problem, considering first the reformulation of Problem 2:

**Problem** **3.**
*Which is the CG-space CG to acquire a cluster of features G considering a set of viewpoint constraints $\tilde{C}$?*


The CG-space represents the multi-dimensional and continuous topological space to capture a feature cluster *G*. As its counterpart, i.e., Cf, the CG holds all valid sensor poses ps to acquire a subset of features *G* considering a set of viewpoint constraints $\tilde{C}$. The present study defines the following terms to formulate CG:The CG-space represents the intersection of all individual feature C-spaces as defined by Equation (Equation 6):
(7)CG=⋂fm∈G(⋂ci∈C~ci(fm,ci)),If a CG-space is a non-empty manifold, i.e., CG ≠ Ø, there exists at least one sensor pose ∃$p_{s}^{C}$ ∈ CG that fulfills all viewpoint constraints to acquire all features *G*.

Figure 5 depicts an abstract representation of the CG-space CG being characterized by the intersection of the individual constrained spaces Cf1 and Cf2 of the feature cluster $\{f_{1},f_{2}\}\in G$.

### 3.2. The Set Cover Problem

CG-spaces provide closed solutions with an infinite set of viewpoints to acquire a feature cluster. However, the main question regarding which features can be acquired from the same viewpoint remains unanswered and is to be investigated within the second subproblem of the VPP, i.e., the SCP. Hence, this section first outlines an adequate formulation of the SCP.

#### 3.2.1. Problem Formulation

In general, the SCP may be formulated as an optimization problem or a decision problem. On the one hand, the optimization definition strives to find the minimal number of *k* viewpoints to capture all *n* features; this formulation was initially introduced in Problem 1. On the other hand, the SCP can be posed as a decision problem:

**Problem** **4.**
*Are k viewpoints sufficient to acquire a set of features F?*


Although any approach addressing the VPP should prioritize the minimization of viewpoints, in our view, an optimization formulation may appear impractical and even ineffective in real applications regarding model uncertainties and feature-rich scenarios. Hence, for the benefit of pragmatism and as other authors, e.g., [15,16,18], have considered, we prefer to relax this requirement and strive for an acceptable number of viewpoints and prioritize robust and computationally efficient solutions.

#### 3.2.2. Solving the SCP

A simple solution for Problem 4 is given by finding *k* iteratively. In this case, a first attempt is made to solve the problem considering an initial value, e.g., k=1. If the *k* viewpoints are not sufficient to acquire *F*, then *k* is increased by one until there are enough viewpoints to acquire all features.

At this point, the challenge and motivation of the first subproblem of the VPP, i.e., the VGP, arises: how the validity of viewpoints can be efficiently and effectively assessed. A solution to this problem was comprehensively addressed using CG-spaces (cf. Section 3.1). Hence, if the number of required viewpoints that fulfill a vision task can be previously approximated, its validation can be efficiently assessed based on CG-spaces.

### 3.3. Reformulation of the VPP

Having addressed the individual subproblems of the VPP, its reformulation can then be posed aligned to a decision formulation of the SCP and the concept CG-spaces:

**Problem** **5.**
*Are k CG-spaces sufficient to acquire a set of features F?*


Aligning the VPP to a decision formulation, the search for finding *k* potential feature clusters must be addressed in the first step. The number of *k* feature clusters can be iteratively found following the ground idea proposed in Section 3.2.2. An adequate strategy to efficiently find feature clusters is comprehensively introduced in Section 4.2.1.

Assuming that an adequate number of potential *k* feature clusters can be found, then exactly *k* CG-spaces must be characterized to capture all features from *F*. Now, if all CG-spaces exist, i.e., ∀{CG1,…,CGk}≠Ø, it can be assumed there exists at least one valid sensor pose within each CG-space. The viewpoint plan that fulfills the regarded vision task can be straightforwardly given by selecting one sensor pose from each CG-space {ps,1,…,ps,k}∈P.

## 4. Methods

### 4.1. Feature Cluster Constrained Spaces

This section describes the definition of CG-spaces based on the intersection of individual C-spaces. First, a strategy to characterize the most elementary C-space based on the sensor’s imaging parameters, feature position, and sensor orientation is introduced. Then, some general terms are established to compute CG-spaces and verify their usability based on an academic example.

### 4.1.1. C-spaces

Aligned to the formulation of C-spaces and treating the VGP as a geometrical problem, [4] demonstrated that a handful of viewpoint constraints can be straightforwardly characterized and integrated using linear algebra, geometric analysis, and CSG Boolean operations.

For instance, the fundamental C-space denoted as C1 is characterized based on the I-space (sensor imaging parameters), a feature of interest and a fixed sensor orientation. Our past study introduced two strategies (extreme viewpoint and homeomorphism interpretation) to characterize the manifold of C1. This paper outlines a simpler variation of the homeomorphism formulation using an alternative reflecting pivot point. The detailed steps to characterize C1 are described in Algorithm 1 and visualized in a simplified 2D representation in Figure 6a–c. Moreover, Figure 6d demonstrates that any sensor pose within C1 is valid to acquire the feature *f*.

The C-space C1 manifold represents the basis for characterizing further C-spaces considering other viewpoint constraints, e.g., the feature geometry, kinematic errors, occlusion, and multi-sensors. The formulation, characterization, and integration of multiple viewpoint constraints fall outside the scope of this publication, see [4] for further details.

**Algorithm 1** Characterization of the C-space C1 by reflection
Select a sensor orientation rsfix for acquiring a feature *f*.Rotate the I-space manifold using the fixed rotation rsfix and let this space be denoted as Is*:
Is*=rotate(Is,rsfix).Reflect Is* over its origin BsIs*:
Is*=reflect(Is*,BsIs*).Position the sensor’s TCP frame at the the feature’s frame considering the sensor orientation rsfix, i.e., $B_{s}^{TCP}=B_{f}$. Let this sensor pose be denoted as
ps0:=ps(ts(BsTCP=Bf),rsfix).Translate the Is* to the sensor lens frame BsIs*=Bss1(ps0). The resulting manifold yields the C-space
Cf1=translate(Is*,Bss1(ps0)).


### 4.1.2. ${}^{G}C$-spaces



**Fixed Sensor Orientation**



Recalling that C-spaces only span a valid solution space for a fixed sensor orientation, it must be assumed that the validity of a CG-space is also limited to a fixed sensor orientation.

Although an approach to characterize C-space regarding multiple orientations was introduced in [4], we consider that in multi-feature scenarios, more efficient solutions can be obtained if the optimization of the sensor is first considered. By doing so, the problem complexity can be reduced for finding a valid sensor position in the Euclidean space considering an optimal sensor orientation.

Although C-spaces and CG-spaces are exclusively valid for a defined sensor orientation, it can be assumed that a deviation of the sensor orientation can be implicitly compensated to some extent within the spanned topological spaces. However, the magnitude of the allowed deviation cannot be explicitly given, and it must be assumed that this is inconsistent within the topological spaces and depends on the remaining constraints.



**Characterization**



Assuming that the sensor orientation is known and the feature clusters are identified, the characterization of the CG-space composed of individual C-spaces is performed in two steps.

In the first step, all individual C-spaces of each feature f∈G are computed considering an individual viewpoint constraint set C˜f with a fixed sensor orientation rsfix. In the second step, the C-spaces of all features belonging to a cluster are merged with each other using a CSG Boolean intersection operation, as formulated in Equation (Equation 7). Figure 7 presents a simplified initial situation, characterization, and validation of a CG-space CG1 for capturing a set of two features $\left\{f_{1},f_{2}\right\}\in G_{1}$.

In addition, Figure 8 considers a more complex scenario in $R^{2}$ for capturing two feature clusters $\left\{f_{1},f_{3}\right\}\in G_{1}$ and $\left\{f_{2},f_{4}\right\}\in G_{2}$. The feature clusters are acquired with the sensor orientations rs,1fix and rs,2fix. The two required CG-spaces CG1(rs,1fix) and CG2(rs,2fix) are characterized by intersecting the corresponding individual C-spaces of the regarded features.

A great deal of attention must be paid if the intersection of two C-spaces yields an empty manifold; it can then be assumed that the corresponding features cannot be acquired simultaneously considering the regarded viewpoint constraints and sensor orientation.



**Verification**



To verify the characterization of CG-spaces for acquiring multiple features, an object comprising four features $\{f_{1},\;f_{2},\;f_{3},\;f_{4}\}\in F$ was designed (see object in the top right of Figure 3). The features are initially grouped into two clusters: $\left\{f_{1},\;f_{2}\right\}\in G_{1}$ and $\left\{f_{3},\;f_{4}\right\}\in G_{2}$. The dimensions and frames of all features are given in Table A3. The features are acquired regarding the following orientation in the object’s coordinate system for the first cluster ros,1 and for the second cluster ros,2:
ros,1(αsz=βsy=0∘,γsx=−150.0∘)ros,2(αsz=90∘,γsx=−160∘,βsy=0∘).

In the first step, the individual C-spaces for each feature were computed according to Algorithm 1 considering the imaging parameters of the sensor $s^{1}$ (see Table A4) and the selected sensor orientations. Moreover, the feature geometry was integrated as a further viewpoint constraint to ensure the acquisition of the entire geometry (see [4]). In the second step, the resulting CG-spaces CG1 and CG2 were synthesized by intersecting the corresponding C-spaces; details on the implementation are given in Section 5.1.4.

Figure 9 shows the described scene and visualizes the characterized manifolds of the individual C-spaces and CGspaces. To verify the validity of the approach outlined within this section, two extreme sensor poses at each CG-space, {ps,1,ps,2}∈CG1 and {ps,3,ps,4}∈CG2, were chosen and rendered, as well as the corresponding depth images and range images at each sensor pose. The results visualized in Figure 10 demonstrate that all features belonging to the same cluster can be simultaneously acquired at the exemplary extreme viewpoints within their corresponding CG-spaces. The academic example comprising the object’s surface model, C-space, CG-spaces, and depth images can be found in the digital appendix of this study.

### 4.1.3. Summary

A C-space represents an analytical, geometric solution offering infinite valid sensor positions to acquire a single feature satisfying a defined set of viewpoint constraints. Moreover, the experiments demonstrated that the intersection of individual C-spaces characterizes a topological space, the CG-space, which provides infinite solutions to acquire a group of features and satisfies all constraints of their C-spaces simultaneously. Due to these characteristics, the present study regards CG-spaces as the backbone element for tackling the VPP.

However, CG-spaces have some limitations, e.g., it must be a priori known which individual C-spaces can be clustered together, their validity is limited to a fixed sensor orientation, and their characterization relies on CSG Boolean operations, which are generally considered an expensive computational technique. Therefore, solving the VPP based on CG-spaces requires a well-thought-out strategy to address these challenges. The formulation of such a strategy is given in Section 4.2.

### 4.2. Viewpoint Planning Strategy

The present publication poses the VPP in the framework of decision formulation problems and suggests the use of CG-spaces as the key element for its solution. This section outlines a holistic viewpoint plan strategy based on CG-spaces to solve generic vision tasks that are confronted with the VPP. Figure 11 provides a simplified overview of the four subordinated modules of the strategy, i.e., the search for potential feature clusters, the optimization of sensor orientation, the characterization of CG-spaces, and the final selection of sensor poses.

#### 4.2.1. Feature Clusters

Addressing the SCP as a decision problem requires identifying feature clusters in the first step, i.e., groups of features that can potentially be acquired together based on their spatial vicinity and orientation similarity. Identifying such feature clusters can be efficiently performed using a clustering algorithm such as a centroid-based k-Means algorithm. This subsection presents a practical and efficient clustering strategy for identifying potential feature clusters. Figure 12 depicts a flow chart of the proposed algorithm.



**Data Preparation**



First, the strategy proposed within this subsection regards a data-enhancing preprocessing step for combining feature positions and their normal vectors to simplify the search for more effective clusters. In addition to the features’ position, we assume that features having similar normal directions are more likely to be clustered together. Hence, a new observation variable is introduced for each feature $f_{m}$, denoted as tfm*∈T*,tfm*∈R3,m=1,…,n, that comprises information about the feature position and orientation. The observation variable is obtained by shifting the features’ frames along their normal (*z*-axis) considering a defined distance $e^{*}$. This trick has the advantage of spatially separating features based on their orientation without increasing the problem’s complexity, see Figure 13. The observation variable tfm* is formally defined as the translation component (trans) of the following transformation:
tf*=transTof(tof,rof)·I00e*01,
where *I* denotes a 3×3 square identity matrix.

Figure 13 depicts a simplified representation of the enhanced variables. The distance $e^{*}$ can be arbitrarily chosen. However, $e^{*}$ should not be larger than the sensor’s far plane $e^{*}\leq h_{s}^{far}$. The empirical experiments in Section 5.1 showed that the sensor’s middle working distance ($e^{*}=d_{s}$) is a good compromise for generating an adequate number of clusters.



**Iteratively Clustering**



Finally, using the set of observations tfm*∈T*, all *k* cluster centroids tj*∈K*, j=1,…,k can be computed using a centroid-based k-Means algorithm. The k-Means aims to choose centroids that minimize the Euclidean distances between a selected cluster centroid and the set of observations $T^{*}$:
∑j=1kmin(‖tfm*−tj*‖2)

Recalling the approach proposed in Section 3.2.2 for iteratively finding a viable number of clusters, a break condition must be first defined to determine when it is necessary to increase the number of feature clusters. Such a condition can be applied for each observation ∀tfm*∈T* considering the minimal Euclidean distance dmin(tfm*,K*) that an observation has to the closest cluster centroid from $K^{*}$. This distance then has to be smaller than a threshold $d_{max}$:
(8)dmin(tfm*,K*)<dmax.

Taking into account the imaging capabilities of the sensor, $d_{max}$ can be defined using the shortest length of the frustum, e.g., half of the sensor width at the nearest view plane (see Figure A1).

If any element of $T^{*}$ does not fulfill condition (Equation 8), then *k* is increased by 1. If a minimal number of clusters can be estimated beforehand, this can be given as an initial value to optimize the search process; otherwise, k=1 should be assumed. This process is repeated until Equation (Equation 8) is satisfied by all observations ∀tfm*∈T*. Figure 13 illustrates the two resulting clusters for a simple scenario in $R^{2}$.

#### 4.2.2. Sensor Orientation Optimization

Recalling the requirements to compute CG-spaces, the present strategy considers a two-step approach for selecting an optimized sensor orientation for each feature cluster.



**Formulation**



The sensor orientation in SO(3) can be fully represented by a normal vector and an optimized swing angle:
rsopt=(nsopt,αsz,opt).

While the normal vector represents the incidence angle, i.e., the rotation around the *x*-axis and *y*-axis of the sensor, the swing angle provides the rotation around the *z*-axis.



**Incidence Angle**



The optimized incidence angle for a cluster can be calculated using the arithmetic mean of the normal vectors of all features of a cluster ∀f∈G, as follows:
(9)nsG,opt=‖∑f∈Gnf‖,nsG,opt<φfsmax.

In the case that the incidence angle between the optimized orientation and a feature is greater than the maximum φfsmax(see Equation (Equation 2)), then the feature must be assigned to a new cluster.



**Swing Angle**



Optimizing the rotation angle around the optical axis can be particularly advantageous when the sensor frustum is asymmetrical. Hence, this subsection suggests finding the optimal swing angle using an oriented minimum bounding box (OBB) algorithm. The direction of the longest side of such a bounding box corresponds to the optimized sensor around its *z*-axis.

First, the 2D projection of all features tf2D∈R2 for ∀f∈G is calculated using a perspective projection matrix denoted as M2D(nsG,opt) and the direction vector nsG,opt:
tf2D=M2D(nsG,opt)·tf.

Next, using all projected points, an OBB algorithm [36] can be applied to compute the four corner points of the bounding box
{g0bb,…,g3bb}←OBB(tf12D,…,tfn2D).

The swing angle corresponds to the principal axis vector eOBBx of the bounding box along the longest side of the bounding box, e.g., $\alpha_{s}^{z,opt}$ is given in the object’s coordinate system (o) as follows:
αosz,opt=arccoseOBBx·eox|eOBBx|·|eoox|,
with eOBBx=g0bb−g1bb. Figure 14 presents an illustrative representation of the identification of bounding boxes and estimation of their orientation used to determine the optimal swing angle for two feature clusters.



**Orientation of further imaging devices**



In the context of range sensors consisting of more than one imaging device (see Section 2.1), note that if the imagingdevices are differently oriented to each other, the orientation for other imaging devices results from the rigid transformation between the devices, e.g., for a second imaging device:
rs2sopt=Rs1s2s·rs1sopt.

The C-spaces for the second device must be characterized using the sensor orientation rs2sopt (see Algorithm 1).

#### 4.2.3. Computation of ${}^{G}C$-spaces



**Integration Strategy of C-spaces**



Having identified the number of necessary feature clusters and corresponding optimized sensor orientations, the necessary CG-spaces can be finally computed. Although the integration of CG-spaces and C-spaces can generally be considered commutative, an adequate computation order of the individual steps can considerably improve the overall computational efficiency of the viewpoint planning strategy.

This study extends the strategy from [4] to consider a more efficient characterization of CG-spaces. Algorithm 2 provides a formal description of these steps. The resulting C-space and CG-spaces manifolds of some of these steps are visualized in Figure 15, considering the academic example introduced in Section 4.1.2. Moreover, it is assumed there exists a kinematic error of 50mm in all directions and that two cubes partly occlude the visibility of the features. The imaging parameters of the second imaging device $s_{2}$ are given in Table A4. We refer to our previous publication, which covered the characterization of kinematic errors, occlusion-free spaces, and integration of viewpoint constraints from multiple imaging devices. The manifolds of all C-space and CG-spaces are found in the digital appendix of this publication.

Moreover, the present strategy considers the decimation of the resulting C-spaces and CG-spaces manifolds by merging adjacent vertices after each CSG Boolean operation to reduce the overall computational effort. Note, that this simplification step may affect the validity of the CG-spaces. Therefore, the threshold value for merging adjacent vertices should be carefully chosen. Some particularities of the overall strategy are addressed in the following subsections.



**Occlusion-Free CG-spaces**



According to [4], C-spaces that integrate occlusion constraints can be characterized using ray-casting and CSG Boolean operations. Therefore, their computation can be regarded as one of the most expensive steps within a viewpoint planning strategy. To optimize the computation of such spaces, the same study showed that the computational efficiency could be considerably improved by limiting the validity of the occlusion-free C-space using the topological space limited by other viewpoint constraints, e.g., imaging parameters, feature geometry, kinematic tolerances.

Based on this insight and taking into account that this step must be repeated multiple times for all C-spaces, the present publication follows a similar approach by exploiting the concept of CG-spaces. Hence, in the first step, the required CG-spaces are computed neglecting any occlusion constraints; let this space be denoted as CGjst,*. In the second step, the individual occlusion-free C-space for each feature can then be more efficiently computed by limiting the occlusion-free visibility to the space spanned by CGjst,* (see Step 6 of Algorithm 2). Note that the same result could have been more inefficiently achieved by first computing the occlusion-free C-space for each feature and then by intersecting all of them. Figure 15 visualizes the resulting occlusion-free CG-spaces following this approach.



**Strategy against invalid CG-spaces**



Finally, it should be noted that the feature clusters and optimized sensor orientations should be considered as an initial and plausible solution to capture all features. However, the validity of a CG-space can be first assessed until the individual C-spaces are intersected and all viewpoint constraints are integrated. Therefore, the existence of the CG-space must be continuously assessed after intersecting two consecutive C-spaces. If the intersection yields an empty manifold or the C-space manifold volume is below a defined threshold, a strategy to overcome this issue must be considered. For instance, in the simplest scenario, the last intersecting C-space can be assigned to a new cluster, and the rest of the process can be continued with the rest of the features.

The intersection of two consecutive C-spaces yielding an empty manifold can be caused by the inconvenient combination of diverse viewpoint constraints. Hence, an efficient strategy to address this issue requires an individual analysis of the particular viewpoint constraint. The comprehensive analysis and formulation of such strategies fall outside the scope of this paper.

**Algorithm 2** Characterization of CG-spaces based on C-spaces
Select the *j* feature cluster Gj∈F, j=1,…,k.Select the *t* imaging device of the sensor st∈S˜, t=1,…,u.Consider the sensor orientation rsts,jopt for the reference imaging device st.Compute all nj C-spaces of all features from the cluster ∀fmj∈Gj, mj=1,…,nj considering the sensor orientation of the first device rsts,jopt and a subset of viewpoint constraints C˜*⊆C˜ that do not require any Boolean operations.
Cfmjst,*:=C(fmj,C˜*,rsts,jopt).Compute the *j* CG-space by intersecting all nj C-spaces iteratively
CGjst,*=⋂njCfmjst,*.Compute all nj occlusion spaces Ofmjst for each feature ∀fmj∈Gj based on the previously corresponding CG-space and the surface models κ∈K of all occluding bodies. The occlusion-free C-space corresponds to the following Boolean difference:
Cfmjst=Cfmjst,*\Ofmjst(CGjst,*,K).Recompute the *j* CG-space iteratively by intersecting all nj occlusion-free C-spaces:
CGjst=⋂njCfmjst.Repeat Steps 3–7 for all imaging devices ∀st∈S˜.Compute the CG-space that integrates the viewpoint constraints of all imaging devices, e.g., for s1, as follows:
CGjS˜,1=CGjs1⋂st∈S˜CGjs1,st(CGjs1,CGjst).Steps 2–11 are repeated for each cluster ∀Gj∈F, j=1,…,k.


#### 4.2.4. Sensor Pose Selection

Having computed all required CG-spaces, the last step considers selecting a sensor pose within each CG-space. Since the present framework does not explicitly consider the existence of a global optimum within a CG-space, any sensor pose within it fulfills all defined constraints and is equally valid to any other; see the depth images in Figure 16.

In the simplest case, any vertex of the CG-space manifold could be used as a valid sensor pose. Alternatively, the geometrical center of the manifold can be considered a local optimum of the CG-space for some cases. However, it cannot be guaranteed that the geometrical center of the CG-space manifold lies within it. For this reason, an explicit evaluation is always required. Such scenarios are not rare when considering occlusion constraints.

### 4.3. Summary

This section outlined a generic viewpoint planning strategy to solve the VPP based on the formulation of the SCP as a decision problem and CG-spaces. First, the strategy uses a k-Means clustering algorithm to identify potential feature clusters. In the second step, a sub-strategy was proposed to consider an optimized sensor orientation in SO(3) for each feature cluster. Having identified potential feature clusters and an adequate sensor orientation, a further generic sub-strategy was introduced to efficiently integrate C-spaces and compute the required CG-spaces manifolds. Finally, some generic approaches were introduced in the last step to select a valid sensor pose within the characterized CG-spaces.

## 5. Results 

This section comprehensively analyzes the usability of CG-spaces and the overall viewpoint planning strategy to solve the VPP in the context of real industrial metrological applications considering two different vision systems. The results confirm that the framework outlined within this publication can be effectively and efficiently used for solving complex vision tasks by providing robust solutions.

### 5.1. Robot Vision System with Structured Light Sensor

#### 5.1.1. System Description

The framework presented within this publication was utilized for automating the dimensional metrology inspection of a car door comprising up to 670 features using the industrial robot vision system AIBox from ZEISS. The AIBox is an integrated industrial RVS manufactured by ZEISS, equipped with a structured light sensor (ZEISS Comet PRO AE), a six-axis industrial robot (Fanuc M20ia), and a rotary table for mounting an inspection object. The imaging parameters of the imaging sensor and structured light projector are given in Table A4. Figure 17 provides an overview of the AIBox and its core elements.

#### 5.1.2. Vision Task Description

To thoroughly evaluate the strategy and algorithms presented in this publication, 15 different inspection tasks were considered. The tasks comprise combinations of different features from both door sides and viewpoint constraints. The left columns of Table A5 provide an overview of the vision tasks.

Door Side: To evaluate the usability of the present framework in an industrial context, a car door was used as the probing object. Due to their topological complexity, feature density, and variability, car doors are well-known benchmark workpieces for evaluating metrology tasks and their automation.Number and Type of Features: The scalability was evaluated using inspection tasks with different numbers and types (points and circles) of features.Viewpoint Constraints: To analyze the efficacy and efficiency of the overall strategy, vision tasks with different viewpoint constraints were designed. All vision tasks regarded at least the most elementary viewpoint constraints $c_{1}$–$c_{3}$ (i.e., the imaging characteristics of the sensor, feature geometry, and the consideration of kinematic errors). Moreover, for some vision tasks, a fourth viewpoint constraint $c_{4}$ was considered to ensure the satisfiability of the viewpoint constraints of the projector. Finally, for the vision tasks that included the viewpoint constraint $c_{5}$, it was assumed that all features must have an occlusion-free visibility to the sensor and projector. Table 1 provides an overview of the considered viewpoint constraints.

#### 5.1.3. Evaluation Metrics

Two metrics were designed to properly evaluate the computed viewpoint plans and to compare and benchmark the obtained results.


**Measurability Index**


To properly quantify the validity of the computed CG-spaces and selected sensor poses for each viewpoint plan, we introduced a measurability index C(P). This metric represents the ratio between the total number of successfully acquired features and the total number of features per view plan *n*. The measurability index is given as follows:
$$M=\frac{\sum g(f_{m})}{n}.$$

To assess the measurability of a single feature, we considered a qualitative $M^{qual}$ and a quantitative $M^{quant}$ metric.

$M^{qual}$:The qualitative function assesses the following two conditions.A feature $f_{m}$, including its entire geometry, must lie within the calculated frustum space Is(ps,j) of the corresponding sensor pose ps,j∈CGjBoth sensor and projector have free sight to the feature.If both conditions are fulfilled, the feature is considered to be successfully acquired.$M^{quant}$:The validity of each feature was further qualified based on the resulting 3D measurement, i.e., the point cloud. This metric counts the number of acquired points within a defined search radius around a feature. If there exist more points than a specified threshold, the feature can be considered to be valid.It needs to be noted that the proper evaluation of this condition requires that the measurements are perfectly aligned in the same coordinate system as the features and that the successful acquisition of surface points is guaranteed if all regarded constraints $c_{1}$–$c_{5}$ are satisfied. Since our work neglects nonspatial constraints that may affect the quality of the measurement (e.g., exposure times or lighting conditions), the validity of the view plans was mainly assessed based on simulated measurements. The simulated measurements are generated by the proprietary software colin3D (Version 3.12) from ZEISS, which considers occlusion and maximal incidence angle constraints. Moreover, the measurements are perfectly aligned to the car door surface model.



**Computational Efficiency**



To provide a comprehensive analysis of the computational efficiency of the viewpoint planning strategy, the computation times of the most relevant steps were estimated:



$t_{k,opt}:$

computation time for computing the necessary *k* feature clusters and corresponding optimized sensor orientations,

tC:

computation time to characterize all individual C-spaces (one for each feature) considering the regarded viewpoint constraints,

tCG:

computation time to characterize all *k* CG-spaces,

$t_{total}:$

total computation time of the vision task, corresponds to the sum of the times mentioned above.

The evaluation tests were developed using the trimesh library [37] and open3D library [38]. All operations were performed on a portable workstation Lenovo W530 running Ubuntu 20.04 with the following specifications: Processor Intel Core i7-3740QM @2.70 GHz, Nvidia Quadro K1000 M, and 24 GB Ram.

#### 5.1.4. Implementation

The viewpoint planning framework was developed based on the Robot Operating System (ROS) [39], which was primarily used for the frame transformation operations. The framework was built upon a knowledge-based, service-oriented architecture. A more detailed overview of the general conceptualization of the architecture and knowledge-base is provided in our previous works [40,41].

Moreover, the clustering was performed using the k-Means algorithm of the Sci-Kit library, using the Lloyd implementation [42]. Due to its high efficiency for performing CSG Boolean operations, the PyMesh library from Zhou et al. [43] was utilized for characterizing the C-spaces and CG-spaces. The ray-casting operations for computing the occlusion space were performed using the trimesh library [37].

#### 5.1.5. Results

In the first step, the required CG-spaces for each vision task of Table A5 were computed using the viewpoint strategy proposed in Section 4.2. An overview of the CG-spaces for the fourth and seventh inspection tasks is displayed in Figure 18. For the feature clustering, we considered a maximal Euclidean distance of $e_{max}=200\;\mathrm{mm}$ (see Equation (Equation 8)), which approximately represents the frustum’s half width-length at its middle plane (see Table A4). Furthermore, the maximal incidence angle was defined as nsmax=30∘. The sensor poses used for the evaluation represented the geometric center of the corresponding CG-space manifolds. The qualitative and quantitative measurability indexes of all viewpoint plans and an overview of the computation times of all vision tasks are shown in the right-handed columns of Table A5. A detailed discussion of the evaluation metrics is discussed as follows.



**Measurability**



In the first step, the qualitative evaluation of all viewpoint plans was performed based on the previously introduced metrics. As expected, the qualitative evaluation shows very encouraging results, with an average measurability index above 95% for the viewpoint plans that considered the most constraints for each inspection task. The high efficacy of some selected viewpoint plans could be further assessed using the quantitative evaluation of simulated and real measurements. Furthermore, it could also be demonstrated that for some single vision tasks (3–5 and 13–15), the measurability index was improved by considering occlusion constraints for the sensor and projector.

On the one hand, these encouraging results confirm the validity and applicability of the viewpoint planning strategy using CG-spaces. In particular, the real measurements for vision tasks 5 and 7 demonstrated that the selected sensor poses were robust enough to compensate for kinematic uncertainties in the real measurements. In contrast, the measurability of some individual features could not be guaranteed in some cases. The failed evaluation of these cases, as well as the discrepancies between the qualitative and quantitative evaluations, can be attributed to two leading causes, which require a further discussion:

Occlusion: All failed qualitative evaluations and the decrease in the measurability score if occlusion constraints were regarded can be attributed to the nonexistence of an occlusion-free space for the computed CG-spaces with the chosen sensor orientation. The strategy proposed in Section 4.2.3 did not explicitly contemplate such cases. However, this problem could be straightforwardly solved by considering an alternative sensor orientation in the 7th step of Algorithm 2 when the intersection of consecutive C-spaces yields a non-empty manifold. Formulating such a strategy requires a more comprehensive analysis of the occlusion space, which falls outside the scope of this work.Furthermore, the failed evaluation of most features lying on the inside of the door was occasioned by occlusion with the door itself, which was initially neglected as an occluding object. However, an empirical analysis of some failed viewpoints showed that a positive evaluation could be achieved by recomputing the CG-spaces of the affected features considering the car door as an occluding object, as seen in Figure 19.Missing points and misalignment: The quantitative evaluation of some individual viewpoints showed discrepancies between the simulation and the real measurements. These differences can be easily explained considering the requirements of the quantitative evaluation strategy proposed in Section 5.1.3 based on the acquisition of surface points. Due to the high reflectivity of the car door material and the fact that the optimization of the exposure time was neglected during the experiments, the acquisition of enough surface points in some areas could not be achieved, see Figure 20. On the other hand, a detailed evaluation of some failed viewpoints showed that the measurements could not be aligned correctly in other cases, causing a false-positive evaluation of some features. By manually optimizing the number of exposure times and individual values, more dense and better-aligned measurements could be obtained, mitigating most of the mentioned errors.



**Computational Efficiency**



The computational efficiency overview from Table A5 demonstrates that all viewpoint plans neglecting occlusion were computed in linear times. Furthermore, the general time distribution for the vision tasks omitting occlusion constraints shows that, on average, 80% of the total time was required for the characterization of the C-spaces manifolds, 15% for the CG-spaces, and only 5% for the clustering and orientation optimization tasks. A more comprehensive analysis of each computed step is discussed below.




$t_{G}:$

It can be observed that the feature clustering and optimization of the sensor orientation can be regarded as the most efficient step of the strategy and represent, on average, less than 10% of the whole planning process. The experiments show the efficiency of the k-Means algorithm for such tasks, agreeing with the previous findings from [28].

tC:

The vision tasks that only incorporate the fundamental constraints $c_{1}$–$c_{3}$ showed a high computational efficiency. These results were to be expected, taking into account that the C-space characterization of these viewpoint constraints consists mainly of linear operations. This trend can be observed in Figure A2, showing the proportional increase between the computation time for the required C-spaces tC and the total number of features. This behavior can be further observed when the fourth constraint is considered. In this case, each C-space must be spanned for each imaging device (sensor and projector), increasing the computation by a factor of two. On the other hand, taking into account the occlusion constraint $c_{5}$ considerably increased the computational complexity of the task. This behavior is also comprehensible, recalling that the characterization of occlusion-free C-spaces relies on ray-casting, which is well known to be a computationally expensive process.Moreover, neglecting occlusion constraints, the average computation time for the characterization of one C-space was estimated at 60 ms. It needs to be noted that this time estimation includes a non-negligible computational overhead of all required operations, such as frame transformation operations using ROS-Services. In [4] the computation of a single C-space was estimated, on average, at 4 ms.

tCG:

The computation of the CG-spaces using intersecting CSG Boolean operations proved to be highly efficient, requiring 10–15% of the total planning time. The experiments also confirm that the time effort increases with the number of intersecting spaces. However, by applying manifold decimation techniques after each Boolean intersection (cf. Section 4.2.3), the time effort could be considerably reduced. For instance, within the first vision task, the characterization of a CG-spacewith six C-spaces required 0.6 s, while the intersection of a CG-spacewith 44 C-spaces took 2.4 s. Furthermore, the computation times of the CG-spaces considering occlusion constraints visualized in Figure A2 confirm that the intersection of more complex manifolds was, on average, more time-consuming.




**Determinism**



Finally, we selected three inspection tasks (4, 6, and 15, marked with * in Table A5) and computed the corresponding viewpoint plan ten times to assess the robustness and determinism of the viewpoint planning strategy. Within these experiments, the heuristic characteristic of the k-Means algorithm can be observed. In particular, within vision task 6, which considers a higher number of features, the number of computed clusters differed between computations with a standard deviation of σ=0.89 clusters. However, the necessary CG-spaces could always be computed, and the computation times showed an acceptable standard deviation.

### 5.2. CMM with Laser Line Scanner

#### 5.2.1. System Description

To assess the transferability of the viewpoint planning strategy and the usability of CG-spaces with an alternative kinematic vision system commonly used in industrial metrology applications, we considered a coordinate measuring machine (CMM) model LK Altera 15.7.6 and a laser line scanner (LLS) LC60Dx from Nikon Metrology. A comprehensive system description, including the synthesis of a digital twin and an exhaustive accuracy analysis, can be found in [44,45]. The CMM can be analogously modeled as a serial robot considering three translational and one rotational degrees of freedom. The imaging parameters of the LLS are found in Table A4.

#### 5.2.2. Vision Task Description

A simple inspection task comprising the dimensional inspection of a self-designed academic probing object consisting of three cylindrical features was regarded to evaluate our framework. The object was coated with a matte white spray to facilitate the acquisition of the cylinders’ surfaces. The measurement system and object are depicted in Figure 21.

#### 5.2.3. Assumptions and Adaptation of the Domain Models and Viewpoint Planning Strategy

Since [4] did not explicitly address the application of C-spaces with LLSs, some additional assumptions regarding the modeling of features, sensors, and viewpoint constraints must be first considered.

Features: The measuring object comprises three cylinders with different positions (see Table A6). The feature frame of the cylinders is placed at the bottom. Taking into account the feature model from Section 2.3, which only considers one frame per feature and assumes that the whole cylinder can be acquired with a single measurement, the definition of the feature model must be extended to guarantee the acquisition of the cylinder’s surface area. Thus, two further features for each cylinder ($f^{+y}$, $f^{-y}$) were introduced. The new feature frames are located at the half-height of the cylinder and their normal vectors (*z*-axis) are perpendicular to the x-axis of the feature’s origin. The geometrical length of these extra features corresponds to the height of the cylinders. An overview of the frames of all features corresponding to one cylinder are shown in Figure 22.Sensors and Acquisition of Surface Points: It is assumed that the LLS only moves in a straight line in the CMM’s workspace with a fixed sensor orientation. For this reason, we assume that all feature surface points can be acquired with a single scanning trajectory as long as the incidence angle constraint between the surface points and the sensor holds.I-space and C-space: Recalling that C-spaces are built based on the 3D sensor’s I-space, we first considered a modification of the LLS’s 2D frustum. Recalling the previous assumption regarding the acquisition of surface points, let the LLS span a 3D I-space composed of the 2D I-space and a width corresponding to the distance of the scanning trajectory. The resulting I-space of one scanning direction and the resulting C-spaces for one cylinder and its three features are visualized in Figure 22. Having characterized a 3D frustum, the approach presented in Algorithm 1 can be directly applied to span the required C-space for one feature. The successful acquisition of one feature results from moving the sensor from an arbitrary viewpoint from one end of the C-space to another arbitrary viewpoint at its other end. This study only considers the characterization of one C-space for the sensor’s laser.C-space and Multi-features: Within multi-feature scenarios, it is desirable to acquire as many features as possible during one linear motion. Therefore, in the simplest scenario, the length of all scanning trajectories corresponds to the size of the object’s longest dimension. Under this premise, we assume that the width of the I-space, hence, of each single C-space, corresponds to this exact length.Viewpoint Constraints: Equally to other vision systems, the successful acquisition of surface points depends on the compliance of some geometric viewpoint constraints, such as the imaging capabilities of the LLS ($c_{1}$) and the consideration of the features’ geometrical dimensions ($c_{2}$). Since the scope of this study prioritizes the transferability of the viewpoint strategy, only these constraints were considered to guarantee the successful acquisition of the regarded features. The adaptation and validation of further viewpoint constraints lie outside the scope of this publication and remain to be further investigated.

#### 5.2.4. Results

Under consideration of the previously mentioned assumptions and modification of the domain models, the necessary CG-spaces were computed to inspect the nine features of the three cylinders aligned to the viewpoint plan strategy proposed in Section 4.2. The clustering and optimization of the sensor orientation can be performed analogously to the presented methods from Section 4.2. However, unlike what is suggested in Section 2.1.3 and assuming that multiple features could be acquired within the same scanning trajectory, a less conservative condition regarding the maximal Euclidean distance of the clustering algorithm was regarded using the maximal length of the workpiece, i.e., $e_{max}=120\;\mathrm{mm}$. Furthermore, the optimization of the swing angle (rotation around the optical *z*-axis, see Section 4.2.2) was particularly useful for estimating the optimized scanning direction of the sensor.

Considering the minimal established viewpoint constraints, the viewpoint planning strategy yielded three CG-spaces to acquire all nine features of the three cylinders of the object. Figure 22 illustrates the resulting CG-spaces manifolds. Each CG-space corresponds to the acquisition of three features for three different acquisition directions (*z*, +y, −y): one at the top (CG1z) and two at both lateral sides (CG2+y, CG3−y) of the object.

The computation time of the whole strategy corresponded to $t_{total}\approx 610\;\mathrm{ms}$, confirming the efficiency of the framework. To validate the computed CG-spaces, four scanning trajectories were selected. Each scanning trajectory’s start and end position corresponds to two extreme CG-spaces vertices, see Table A5. Figure 23 visualizes the corresponding scanning trajectories of all CG-spaces.

Furthermore, an exemplary representation of the sensor placement of the resulting scanning trajectories of the third CG-space CG3−y, the corresponding 2D frustum, and the acquired surface points are shown in Figure A3. These extreme scanning trajectories demonstrate the validity of the computed CG-space, which guarantees that the height of all three cylinders and the outer radii always lie inside the I-space for all four scans. It can also be assumed that any arbitrary combination of start and end positions within the outer planes of the CG-spaces will also be valid. The qualitative measurability function from Section 5.1.3 could be straightforwardly applied to all scanning trajectories and corresponded to the complete acquisition of all features for at least one trajectory.

It should be noted that some surface points could not be successfully acquired for some single extreme scanning trajectories, e.g., the scans 1 and 2 from Figure A3. This failed acquisition can be attributed to the manufacturing tolerances of the self-designed object. However, such tolerances can be seamlessly compensated by selecting alternative sensor positions within the CG-spaces or considering tolerances implicitly in the synthesis of the C-spaces, as in the first evaluation case (cf. $c_{3}$ from Table 1).

Furthermore, the measurements of all scanning trajectories are depicted in Figure 24, confirming the validity of the viewpoint plan and computed CG-spaces. The surface model of the object, manifolds of the computed CG-spaces, and the performed measurements can be found in the digital appendix of this paper. From these experimental results, it can be concluded that the proposed viewpoint planning strategy based on C-spaces and CG-spaces could be satisfactorily applied for vision systems comprising LLSs under consideration of certain assumptions.

### 5.3. Discussion

A more comprehensive evaluation of the strengths and limitations of the viewpoint planning strategy is discussed in the context of the main goals of this publication:

#### 5.3.1. Efficacy

On the one hand, the effectiveness of the computed viewpoint plans and chosen sensor poses showed an encouraging average measurability satisfiability of over 95% for all regarded vision tasks. It was demonstrated that by considering more viewpoint constraints, the efficacy of the viewpoint plans could be increased towards full measurability. On the other hand, in some cases, a complete measurability could not be achieved due to the occlusion limitations of the planning strategy. Although the present work considered occlusion constraints and the validity of occlusion-free C-spaces could be verified for individual cases, a proper strategy to consider more complex scenarios remains a stimulus for further research.

Moreover, although the CG-spaces are valid only for a fixed sensor orientation, the experiments showed that the selected sensor poses were also effective and robust for compensating for minor orientation deviations. In case an orientation range must be explicitly considered, the future works should then extend the strategy to consider CG-spaces for multiple sensor orientations, as suggested in [4].

Furthermore, most of the selected viewpoints within the CG-spaces were shown to be valid and sufficient for passing the considered evaluation metrics. However, within the evaluation, we also observed that some real measurements differed from the simulated evaluations due to modeling uncertainties or manufacturing tolerances. An individual analysis of some of these measurements demonstrated that these uncertainties could be compensated for by selecting an alternative sensor pose within the corresponding CG-space. This characteristic of CG-spaces embodies the intrinsic benefits of C-spaces for compensating for uncertainties without compromising the validity of the viewpoint plan. However, these findings also suggest that the further research should be devoted to designing optimization strategies for finding alternative viewpoints within CG-spaces to compensate for target-oriented uncertainties and neglected constraints.

#### 5.3.2. Computational Efficiency

The comprehensive analysis of the computational complexity demonstrated that most view plans could be computed in near linear time. Due to the complexity of the developed software framework and dependency on external libraries, a more detailed computational analysis is needed to verify the complexity of the strategy and C-space. However, our experiments confirmed the strength and simplicity of CG-spaces, showing that effective viewpoint plans could be computed within feasible times despite the complexity of the vision tasks and vision systems.

Moreover, the estimated computational times also showed that if occlusion constraints are regarded, the complexity of the overall strategy increases considerably and depends strongly on the complexity of the considered surface models. Hence, the future research should be devoted to a more efficient characterization and computation of the occlusion spaces.

Furthermore, in terms of overall planning efficiency, the further studies should concentrate on the extension of the present viewpoint planning strategy for considering the proper combination of CG-spaces in case of overlapping (e.g., see right-handed image of Figure 18) to reduce the number of required viewpoints.

#### 5.3.3. Transferability

This study demonstrated that the usability of C-spaces, CG-spaces, and the overall viewpoint planning strategy could be satisfactorily evaluated with different vision systems. In the context of a simplified inspection scenario and considering an adaption of the models introduced in the current study, the applicability of the suggested strategy and potential of CG-spaces for LLSs was demonstrated.

However, further work remains to be carried out to properly evaluate the use of CG-spaces for LLS and to consider more complex vision tasks comprising further viewpoint constraints, e.g., occlusion constraints.

## 6. Conclusions

### 6.1. Summary

The VPP is a multi-dimensional and challenging problem that any vision task demanding the computation of multiple and valid viewpoints must consider. Based on our literature review, the VPP is still regarded as a complex and unsolved challenge, lacking a generic and efficient framework for computing multiple viewpoints within feature-based applications.

Towards tackling the VPP, the present work proposes its modularization and addresses its subproblems separately, i.e., the VGP and the SCP. First, based on our previous work [4], the present study poses the VGP for multi-feature applications, addressing it as a purely geometric problem that can be solved based on CG-spaces. CG-spaces span continuous solution spaces in 6D, providing an infinite number of valid viewpoints that guarantee the successful acquisition of feature clusters, taking into account various viewpoint constraints. The experiments undertaken within this publication demonstrate that spanning an infinite solution space is a powerful technique for implicitly considering model uncertainties within real applications and delivering alternative solutions. Moreover, to address the SCP, we proposed a holistic viewpoint planning strategy based on a heuristic clustering method to identify the sufficient number of CG-spaces required to fulfill a vision task. The validity, efficiency, and transferability of the viewpoint planning strategy proposed in this study were evaluated using two different industrial vision systems within dimensional metrology tasks. Our evaluation showed that valid viewpoints could be computed for diverse inspection tasks and sensors in linear times, guaranteeing an acquisition of up to 90% of all features.

The key contributions and advantages of a viewpoint planning strategy based on CG-spaces are summarized as follows:

Mathematical and generic formulation of the VPP to ease the transferability and promote the extensibility of the framework for diverse vision systems and tasks.Synthesis of CG-spaces built upon C-spaces, inheriting some of their intrinsic advantages:-analytical, model-based, and closed-form solutions,-simple characterization based on constructive solid geometry (CSG) Boolean techniques,-infinite solutions for the seamless compensation of model uncertainties.Generic and modular viewpoint planning strategy, which can be adapted to diverse vision tasks, systems, and constraints.

### 6.2. Limitations and Future Work

The outlined viewpoint strategy can be categorized as a model-based approach requiring minimal a priori knowledge about the sensor’s frustum model and the location and geometry of the features to be acquired. However, the present framework does not consider a stringent definition of features. Hence, the future work should evaluate its usability and adaptability for diverse vision tasks.

Our previous work [4] introduced the core concepts required for synthesizing CG-spaces and demonstrated that several viewpoint constraints could be modeled geometrically. However, in the context of applications demanding multiple viewpoints, we observed that the potential of CG-space remains to be further exploited for considering further constraints. On the one hand, based on our ongoing research and preliminary results, we still see the potential for explicitly characterizing registration constraints for maximizing and ensuring the overlapping area between adjacent measurements [46]. On the other hand, we can also imagine that CG-spaces could be used for considering further robot constraints such as sensor lighting parameters, robot collisions, cycle times, and energy-efficiency constraints. Concretely, our current research concentrates on exploiting the use of CG-spaces for simultaneously optimizing the sensor exposure time and sensor pose.

Although our experimental results demonstrated encouraging results regarding the efficacy and efficiency of the present framework, further studies should concentrate on the definition of a benchmark scenario and standardized evaluation metrics that facilitate a direct and more comprehensive evaluation between diverse approaches. Due to the nature of the approach presented, which suggests an explicit characterization of the domain models and constraints, a non-negligible effort to implement and parameterize the models as proposed in this research should be considered.

Despite the mentioned limitations, we are convinced that the outlined viewpoint planning strategy based on CG-spaces provides a springboard for a novel and efficient approach for tackling the VPP, comprising closed and deterministic solutions. We hope that our findings aid researchers and industry in enabling the automation of diverse vision tasks.

## Figures and Tables

**Figure 1 sensors-23-07964-f001:**
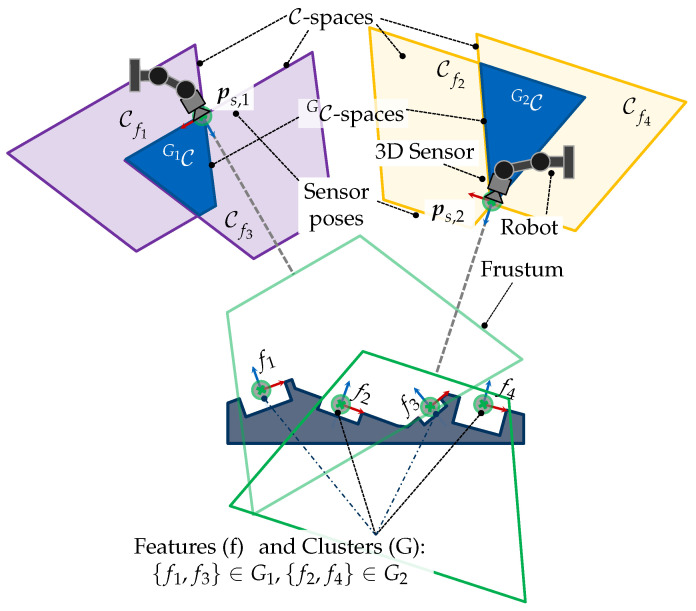
Simplified, graphical representation of the Viewpoint Planning Problem (VPP): how many viewpoints (sensor poses) are needed to acquire all four features? The present study proposes a viewpoint planning strategy based on feature cluster constrained spaces (CG-spaces) to answer this question. A CG-space spans a topological space in the special Euclidean SE(3) with an infinite number of sensor poses ∀ps∈CG to acquire all features from a cluster *G* that satisfy a set of viewpoint constraints $\tilde{C}$, e.g., imaging parameters of the sensor, feature geometry, and orientation. CG-spaces are computed based on the intersection of C-spaces to acquire individual features. This example shows that two sensor poses (ps,1, ps,2) are required to capture the four visualized features. The selection of the sensor poses is performed straightforwardly by selecting any sensor pose within the CG-spaces CG1 and CG2. The design of a strategy for selecting which features can be grouped and the characterization of the CG-spaces are the focus of the present research.

**Figure 2 sensors-23-07964-f002:**
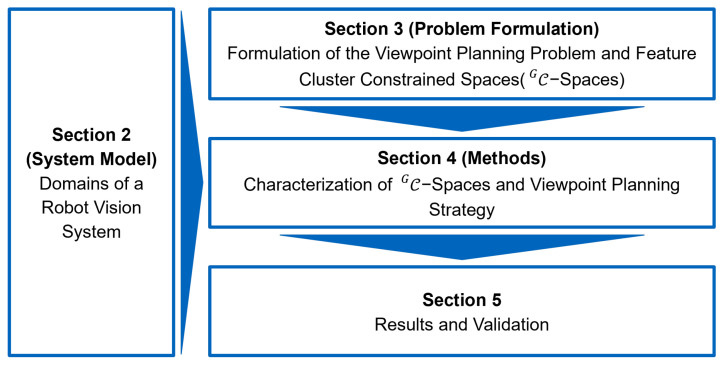
Outline.

**Figure 3 sensors-23-07964-f003:**
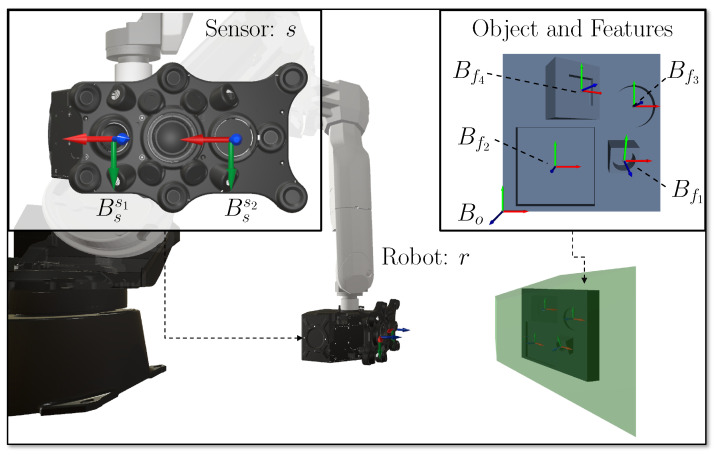
Overview of the most relevant components of a robot vision system.

**Figure 4 sensors-23-07964-f004:**
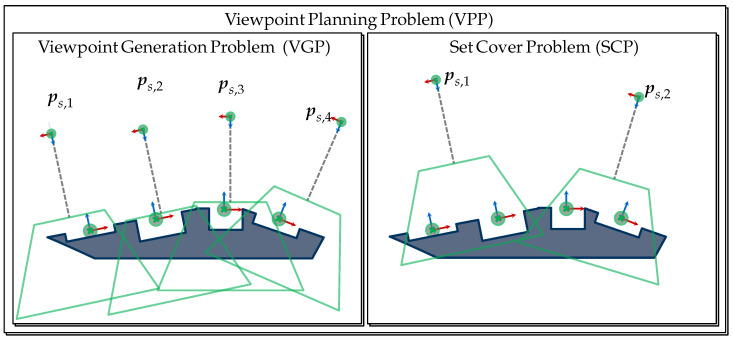
Modularization of the VPP and simplified representation of its subproblems. On the one hand, the VGP addresses the acquisition of a single feature by a viewpoint satisfying a set of constraints. On the other hand, the SCP seeks to reduce the number of required viewpoints to acquire all features.

**Figure 5 sensors-23-07964-f005:**
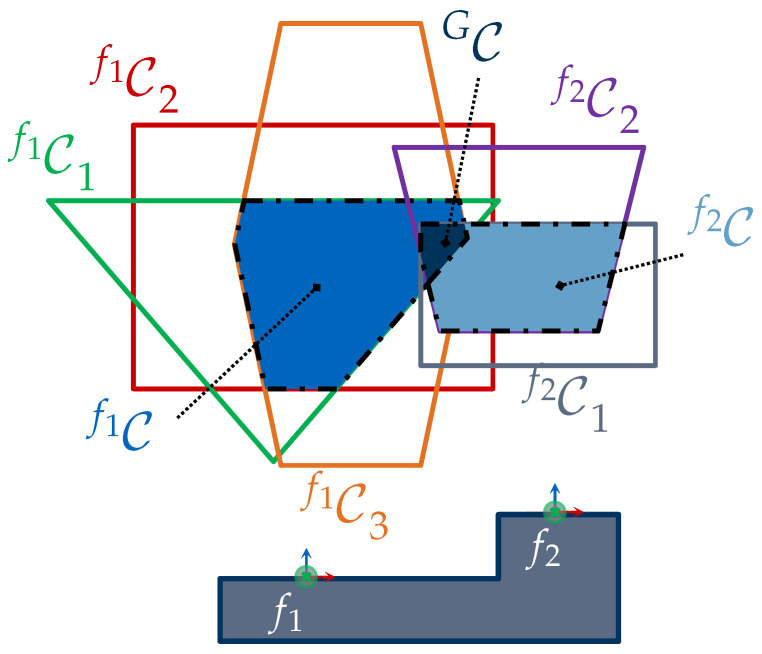
Abstract representation of the C-spaces Cf1 and Cf2 that represent the infinite solution space for separately acquiring the features $\{f_{1},f_{2}\}\in G$. Cf1 and Cf2 are characterized by different viewpoint constraints and their respective C-spaces {Cf11,Cf12,Cf13}⊇Cf1 and {Cf21,Cf22}⊇Cf2. The intersection of the two C-spaces Cf1 and Cf2 yield the CG-space CG, which represents the infinite solution space to simultaneously acquire all features of the feature cluster *G*, fulfilling all viewpoint constraints.

**Figure 6 sensors-23-07964-f006:**
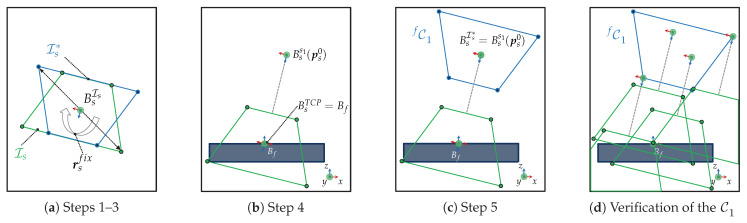
Characterization and Verification of the core C-space C1 for capturing feature *f* considering a fixed sensor orientation and the I-space. (**a**–**c**): Simplified visualization of the steps of Algorithm 1 to characterize the C-space C1, which considers the imaging parameters, the feature position, and a fixed sensor orientation. (**d**): Any sensor pose with ps(rs=rsfix)∈C1 is valid to capture feature *f*.

**Figure 7 sensors-23-07964-f007:**
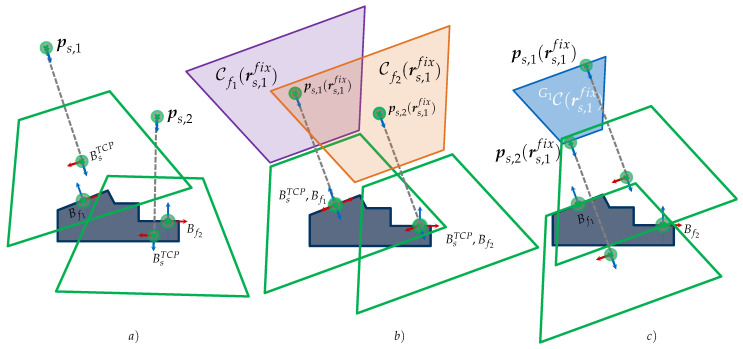
Overview for characterizing CG-spaces. (**a**) Initial situation: Two features $f_{1}$ and $f_{2}$ with two corresponding sensor poses ps,1 and ps,2 to capture them. (**b**) Step 1 for CG-space characterization: Select a sensor orientation rs,1fix for capturing both features and estimate their corresponding C-spaces Cf1 and Cf2. (**c**) Step 2 for CG-space characterization and verification: Intersect both C-spaces to characterize the resulting CG-space: CG1=Cf1⋂Cf2. Any sensor pose within ∀ps∈CG1 is valid for capturing both features.

**Figure 8 sensors-23-07964-f008:**
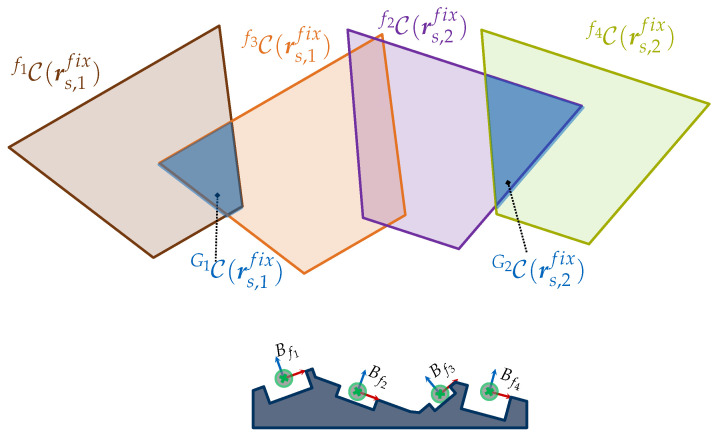
Simplified representation of two CG-spaces CG1(rs,1fix) and CG2(rs,2fix) to acquire the feature clusters $\left\{f_{1},f_{3}\right\}\in G_{1}$ and $\left\{f_{2},f_{4}\right\}\in G_{2}$.

**Figure 9 sensors-23-07964-f009:**
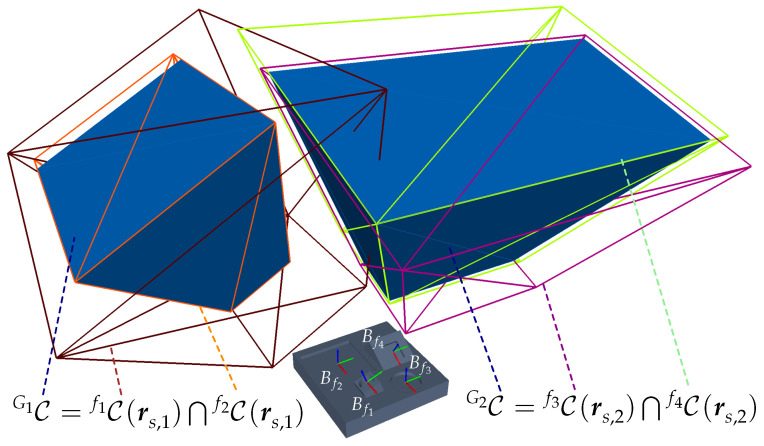
Manifolds of the CG-spaces CG1 and CG2 in SE(3) for two feature clusters $\left\{f_{1},\;f_{2}\right\}\in G_{1}$ and $\left\{f_{3},\;f_{4}\right\}\in G_{2}$ being characterized by the intersection of the corresponding C-spaces Cf1, Cf2, Cf3 and Cf4.

**Figure 10 sensors-23-07964-f010:**
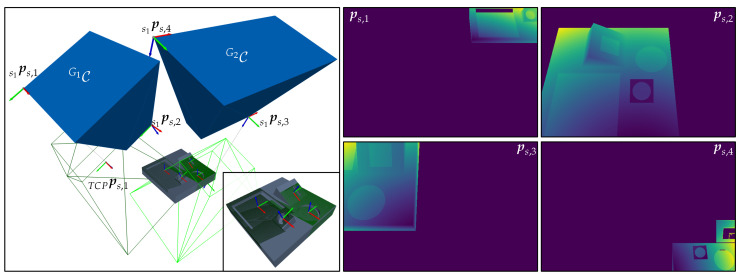
Verification of CG-spaces considering two sensor poses {ps,1, ps,2}∈CG1 and {ps,3, ps,4}∈CG2 at the vertices of each manifold. Rendered scene and range images of ps,1 and ps,3 of (**left** image) and depth images of all sensor poses (**right** images).

**Figure 11 sensors-23-07964-f011:**
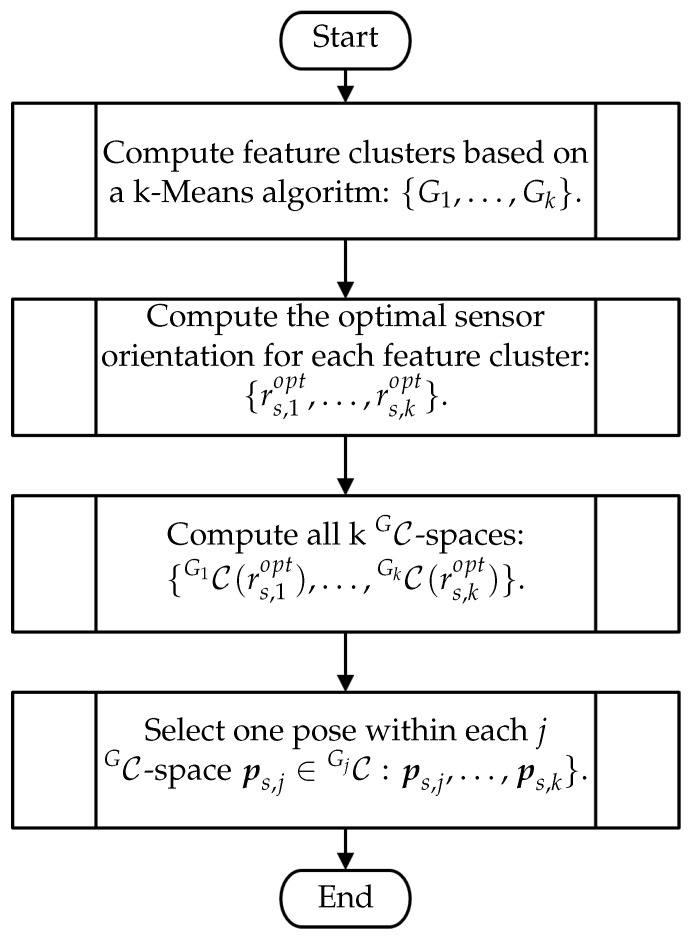
Overview of the viewpoint planning strategy main modules.

**Figure 12 sensors-23-07964-f012:**
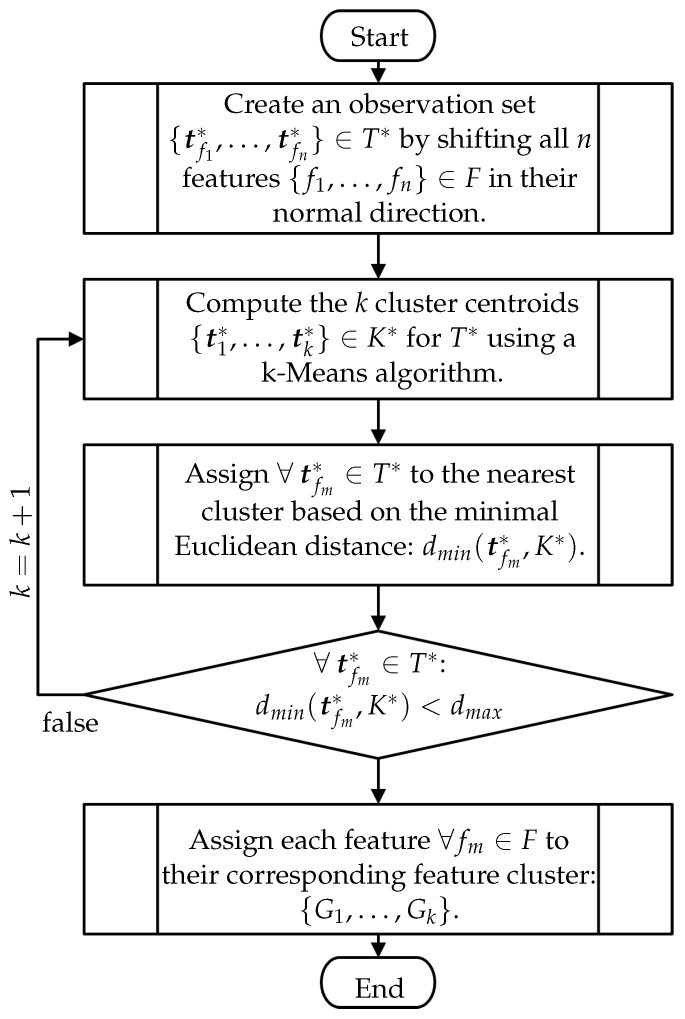
Algorithm for computing feature clusters based on a k-Means algorithm.

**Figure 13 sensors-23-07964-f013:**
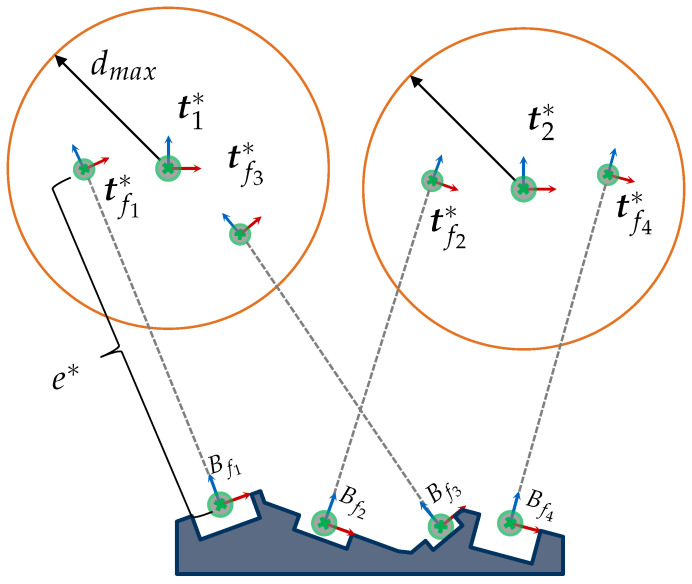
Exemplary characterization of two feature clusters (centroids: t1*,t2*) considering the observation set tf1*,…,tf5* using a k-Means algorithm.

**Figure 14 sensors-23-07964-f014:**
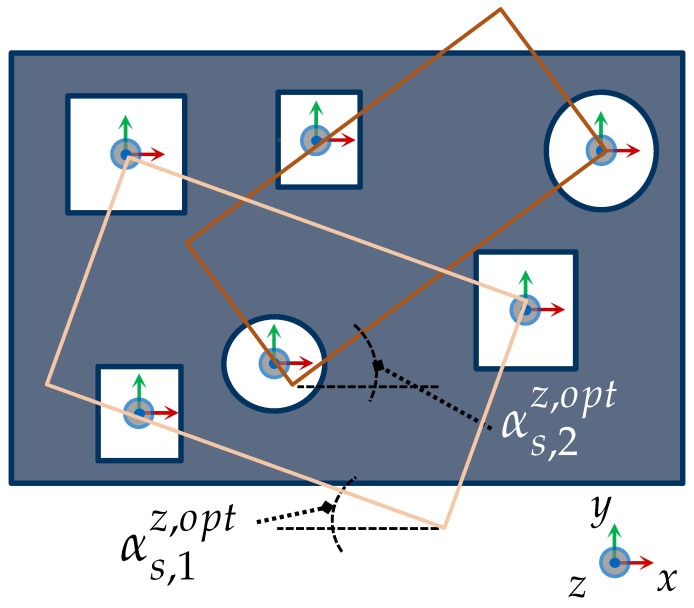
Exemplary optimization of the swing angles $\alpha_{s,1}^{z,opt}$ and $\alpha_{s,2}^{z,opt}$ for two feature clusters using an OBB algorithm.

**Figure 15 sensors-23-07964-f015:**
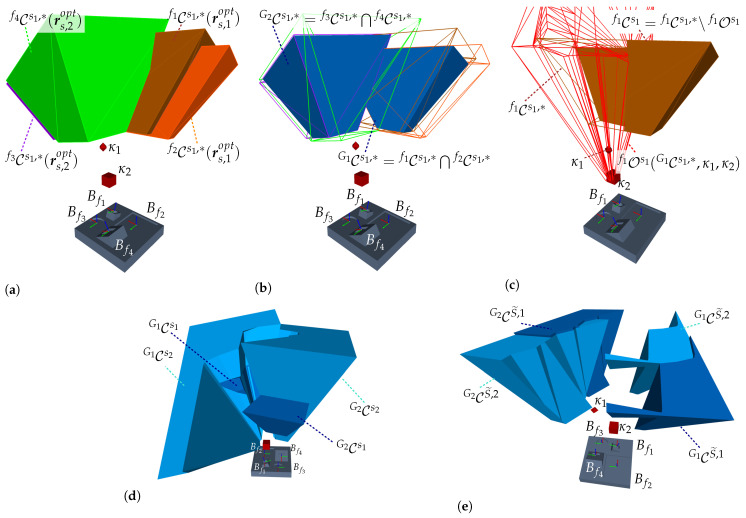
(**a**–**e**): Exemplary visualization of some steps of Algorithm 2 to characterize the CG-space based on three individual C-spaces corresponding to three features. (**a**) Step 4: Compute the C-spaces for $s_{1}$ of all features considering the optimized sensor orientations rs,1opt for $\left\{f_{1},f_{2}\right\}\in G_{1}$ and rs,2opt for $\left\{f_{3},f_{4}\right\}\in G_{2}$. (**b**) Step 5: Compute the CG-spaces CG1 and CG2 by intersecting the corresponding C-spaces. (**c**) Step 6: The occlusion-free C-space for $f_{1}$ is characterized by computing the Boolean difference between the occluding space Of1st(red wireframe manifold) and the C-space Cf1s1,* from Step 4. (**d**) Step 7: Compute the occlusion-free CG-spaces for the first imaging device CG1s1 and CG2s1. The CG-spaces of the second imaging device $s_{2}$ (CG1s2, CG2s2) are computed analogously following Steps 3–7. (**e**) Step 9: Compute the CG-spaces for $s_{1}$ CG1S˜,1 and CG2S˜,1 to consider the viewpoint constraints of the second imaging device.

**Figure 16 sensors-23-07964-f016:**
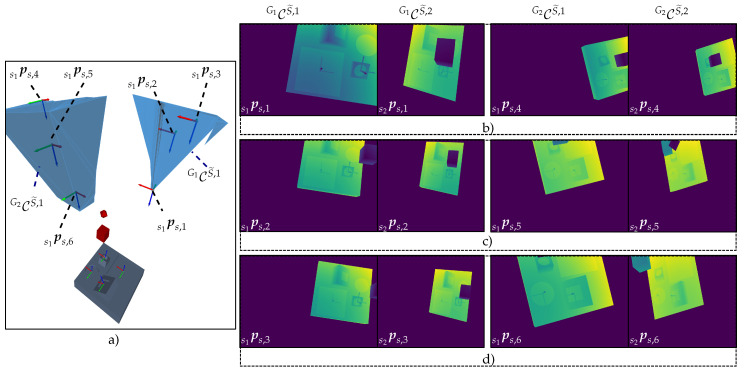
(**a**) Verification scene comprising two CG-spaces for acquiring two feature clusters $G_{1}$ and $G_{2}$ and three potential sensor poses within each CG-space. The images on the right depict the rendered depth images of the imaging devices $s_{1}$ and $s_{2}$ corresponding to the three different sensor poses within the CG-spaces at (**b**) a vertex, (**c**) the geometric center, and (**d**) one random point.

**Figure 17 sensors-23-07964-f017:**
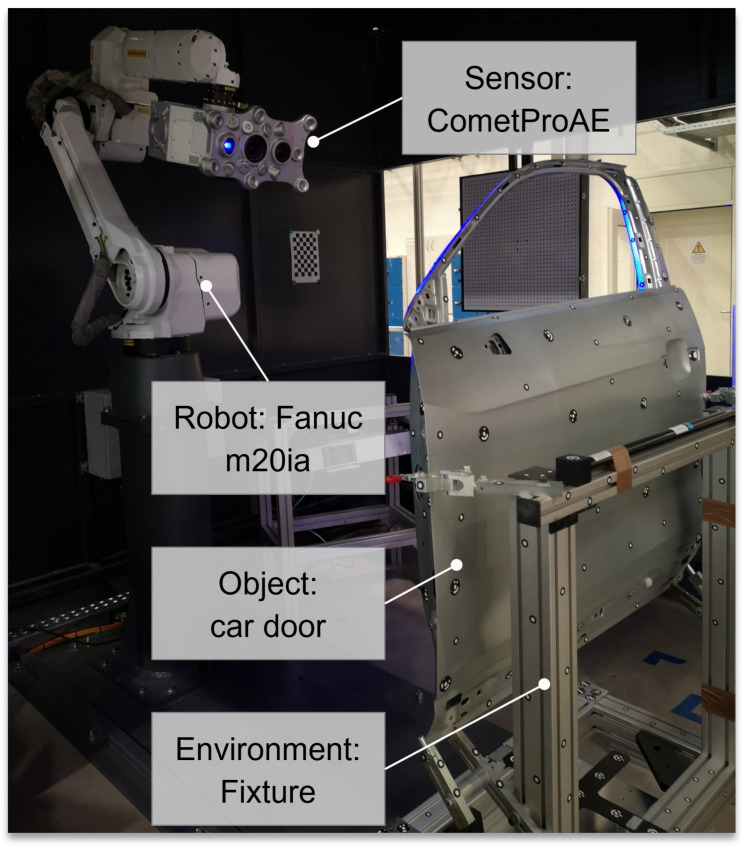
Overview of the core components of the *AIBox*.

**Figure 18 sensors-23-07964-f018:**
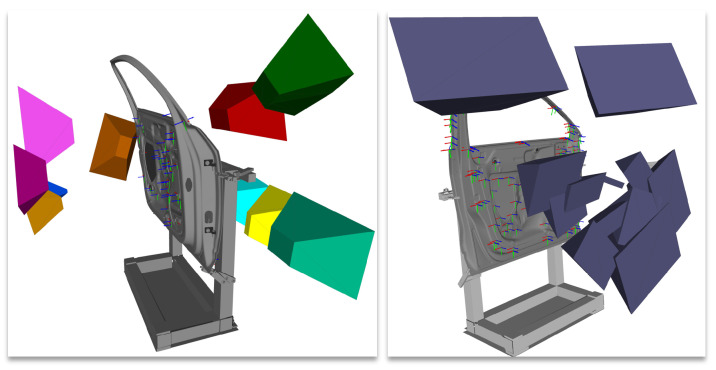
CG-spaces of inspection tasks 4 (**left**) and 7 (**right**).

**Figure 19 sensors-23-07964-f019:**
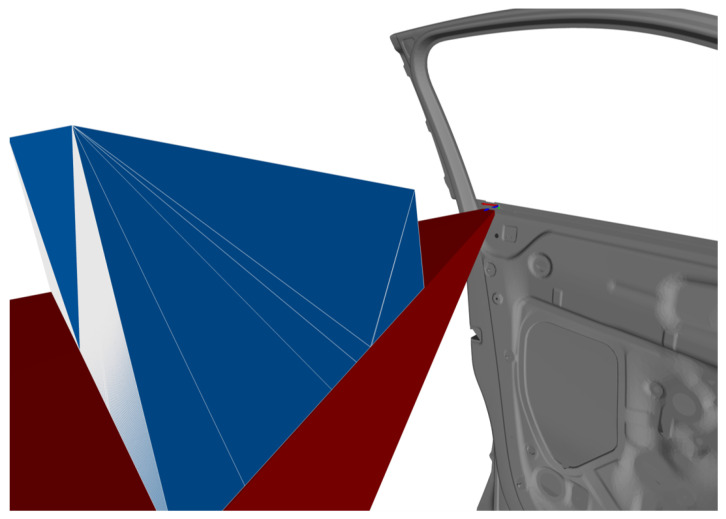
Recomputation of CG-space regarding the door as an occlusion object, the red manifold represents the occlusion space generated by the door’s surface model.

**Figure 20 sensors-23-07964-f020:**
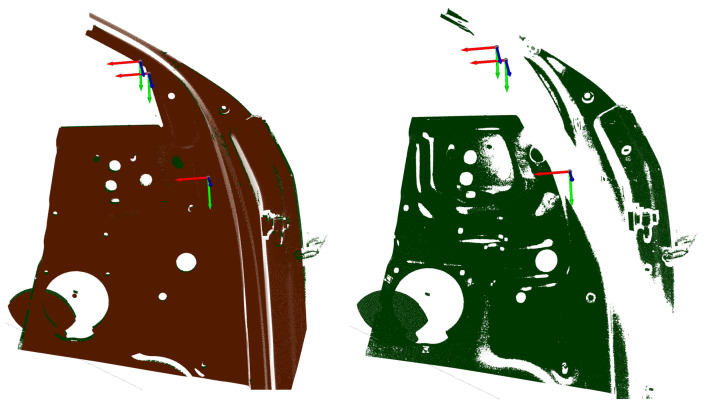
Comparison of the quantitative evaluation of the measurability of three exemplary features using a simulated measurement (**left**) and a real measurement (**right**): the quantitative assessment with real measurements fails due to a lack of surface points.

**Figure 21 sensors-23-07964-f021:**
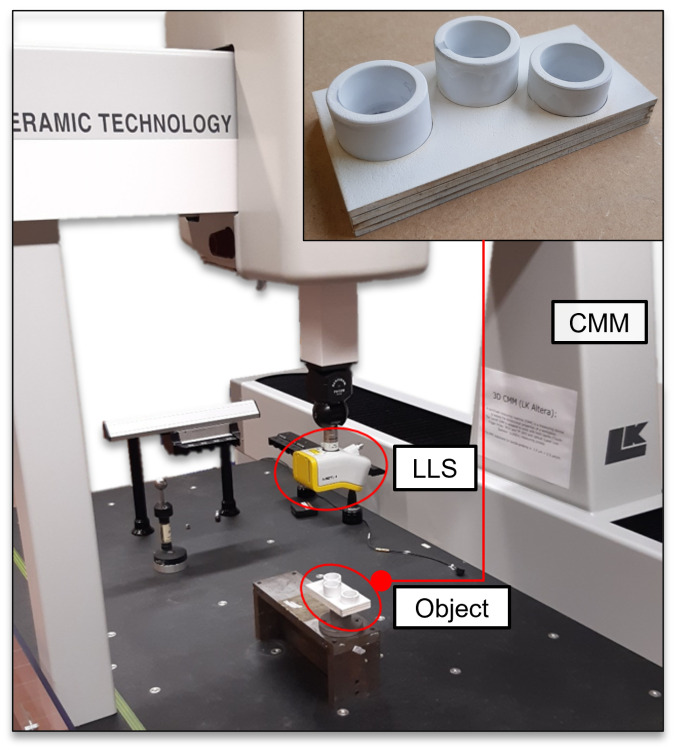
A CMM vision system consisting of two main hardware components: a CMM and the LLS. The probing object composed of three cylinders is positioned within the workspace of the CMM.

**Figure 22 sensors-23-07964-f022:**
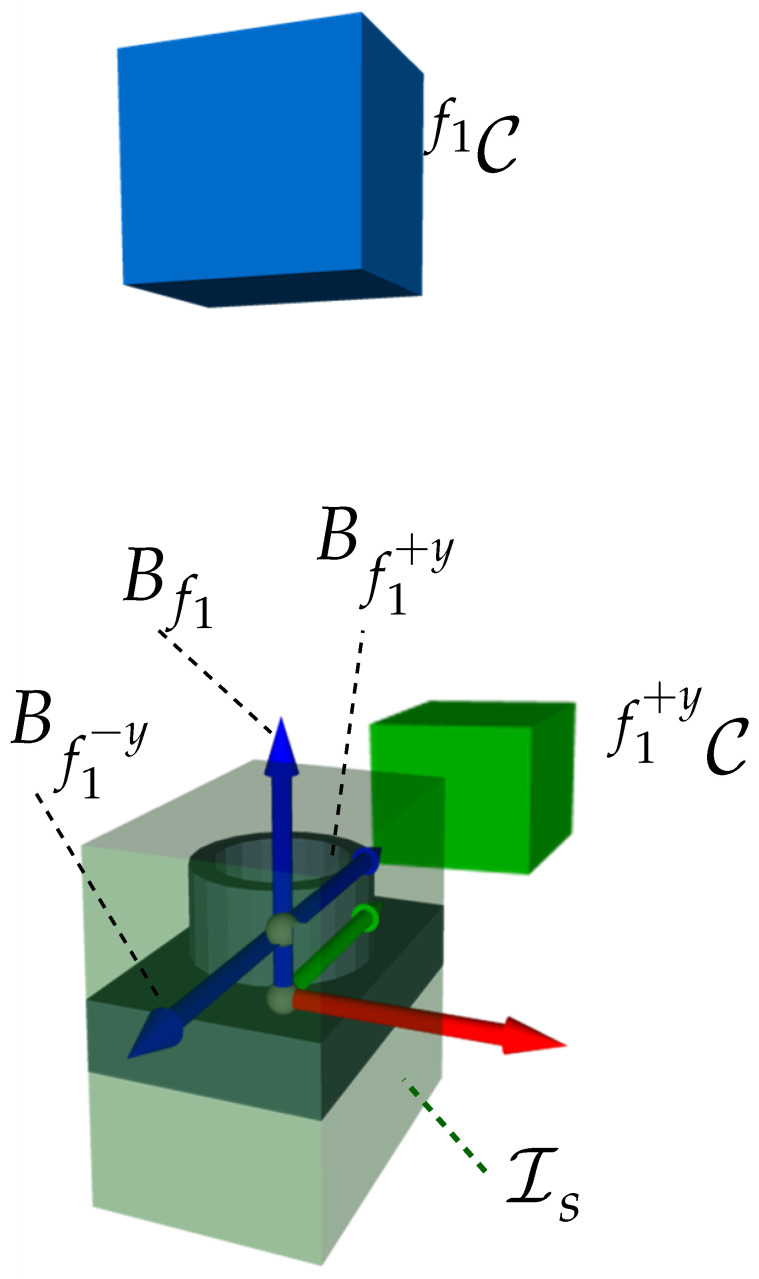
Cylinder with three features, extended model of the sensor I-space (Is) for an LLS, and two exemplary C-spaces for each feature $f_{1}$ and $f_{1}^{+y}$.

**Figure 23 sensors-23-07964-f023:**
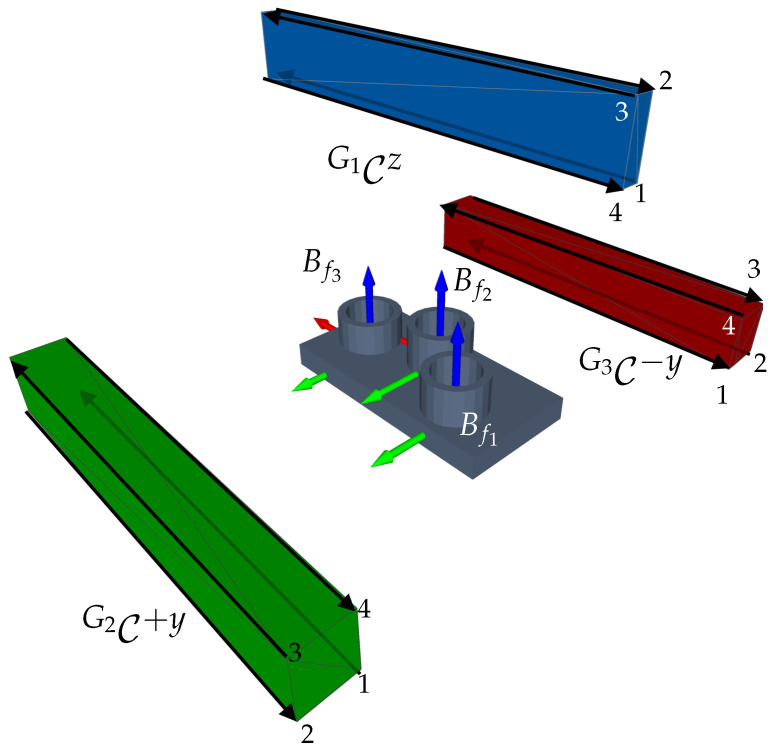
Overview of the three CG-spaces (CG1z, CG2+y, CG3−y) and visualization of the four scanning tracks (black lines).

**Figure 24 sensors-23-07964-f024:**
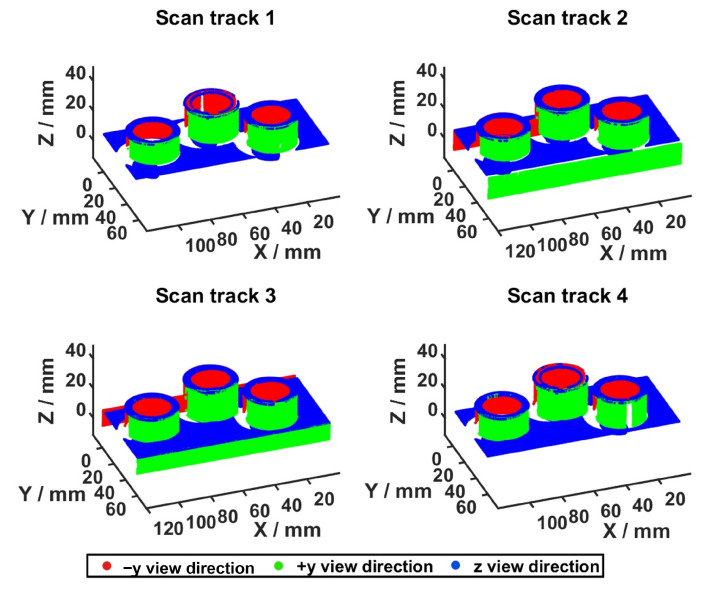
The combined point cloud of the three view directions for all four scan tracks.

**Table 1 sensors-23-07964-t001:** Overview of viewpoint constraints.

Viewpoint Constraints	Description
$c_{1}$	The imaging parameters of the structured light sensor comprising the camera and projector must be considered (see Table A4).
$c_{2}$	The feature dimensions must be regarded so the whole feature geometry is acquired within the same measurement.
$c_{3}$	Due to modeling uncertainties, a kinematic error of 50mm in all Cartesian directions is assumed.
$c_{4}$	The imaging parameters of the structured-light projector must be considered.
$c_{5}$	The door fixture may occlude some features. However, a self-occlusion with the car door is neglected.

## Data Availability

A comprehensive set of Appendix B is attached to this paper. This set includes CG-spaces and C-spaces meshes, meshes and frames of the used solid models, features and viewpoint frames, and the rendered data used for verification and validation.

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
