# Peer review of "Viewpoint Planning for Range Sensors Using Feature Cluster Constrained Spaces for Robot Vision Systems"

_sensors, 2023, doi:10.3390/s23187964_

Round 1
Reviewer 1 Report
In this manuscript, Magana and coworkers outlined the computation of valid viewpoints as a geometrical problem and proposed Feature-Based Constrained Spaces (C-spaces) to tackle this problem efficiently for acquiring one feature. Then they extended the concept of C-spaces to consider multi-feature problems using a new version of model (GC-spaces). The current study also outlined a generic viewpoint planning strategy for solving vision tasks comprising multi-feature scenarios. They finally validated the framework on two different industrial vision systems used for dimensional metrology tasks. In summary, this manuscript was well-written, very comprehensive, and the experiments were very carefully designed and performed. All the backend data was also provided in detail. The significance and novelty were clearly stated that the outlined viewpoint planning strategy based on GC-spaces provides a springboard for a novel and efficient approach for rackling the VPP, comprising closed and deterministic solutions. After all, I think this paper can be published with the current form.
Comments:
1) In line 236, the authors mentioned that “the present study considered robots to be an optional element of a vision system”. Why are robotic movements not important? I personally think real-time vision from robotic arm and x, y, z, rx, ry, rz (r: rotational)-axes movements of industrial robot are critical for study of vision. Please explain more. Thanks!

Author Response
The authors are grateful for the reviewer's helpful comments and feedback.
Answer to your observation:
The robot is definitely a critical component to enable the automation of such applications. However, since we assume that the images are captured statically, the presented framework considers a robot to be simply a positioning device for the sensor. For this reason, in the current scope of the framework, we neglect any dynamic properties that may affect the acquisition of measurements. The only limiting constraint is the robot's workspace, which was considered in our previous work. Moreover, in the validation, we showed that the framework could be extended to other positioning devices, such as the CMM, without considering further constraints.
Reviewer 2 Report
Viewpoint Planning is a high-dimensional optimization problem that is difficult to solve. The methods proposed in the manuscript are effective and informative, which have significant engineering application value. I think the manuscript meets the publishing standards. The following comments can be considered:
(1) The readability of the manuscript can be further improved, and the organization of materials can be simplified;
(2) The format of references lacks standardization.
Author Response
The authors are grateful for the reviewer's helpful comments and feedback.
Answer to your observations:
1) We recognize that the structure of the paper does not follow the standard outline of other publications. Based on internal discussions and feedback from previous reviewers, we have decided that the current structure best fits the broad content of the paper.
2) Citations have been standardized in the latest version.
Reviewer 3 Report
Here is a detailed review of the paper with suggestions for major revisions, without the references:
Major Revisions:
1. The introduction provides a good high-level overview of the viewpoint planning problem and past work. However, it could be expanded to more clearly motivate the limitations of existing methods that this work aims to address. What are the key gaps in prior art that the proposed approach fills? Some more discussion is needed.
2. The concept of G-C spaces needs clearer explanation early on. A figure or diagram showing the intersection of individual C-spaces to create a G-C space would help visualize this better. How exactly does a G-C space extend the representation to handle multiple features? Walk through a simple example.
3. The formulation of the viewpoint planning problem using G-C spaces in Section 3 requires better description. Are G-C spaces computed per feature cluster? How do the constraints and thus the G-C space differ across clusters? More details are needed on how individual C-spaces are combined into a G-C space.
4. In Section 4.1, an example walking through the steps to compute a G-C space for a small feature cluster would make the process much clearer. Show the individual C-spaces and their intersection visually.
5. Explain the clustering algorithm for identifying potential feature groups in more detail. What objective function or criteria is used? How are cluster compactness and number of clusters K determined?
6. The occlusion handling could be compared to recent methods that also use visibility computations. How does the proposed occlusion handling relate?
7. The experimental results demonstrate the framework well, but need more analysis. Can performance be compared to a baseline or brute force method? How does computation time and accuracy vary with number of features and constraints?
8. For reproducibility and comparison, provide more details on dataset characteristics, error metrics used, training procedures and hyperparameters.
In the section discussing related work, it would be beneficial to include a review of some closely related works, such as:
"Robust Feature Matching for Remote Sensing Image Registration via Guided Hyperplane Fitting" (TGRS22)
"PGFNet: Preference-Guided Filtering Network for Two-View Correspondence Learning" (TIP 23)
In summary, I think the core ideas proposed are interesting but require much clearer explanation, especially on the G-C space computation and clustering. More implementation details and experimental analysis also will strengthen the paper. Comparisons to recent methods will also help highlight the benefits. With major revisions along these lines, the paper can become a stronger contribution. Please let me know if any of these suggestions need clarification!
see above
Round 2
Reviewer 3 Report
After reviewing this manuscript, I believe the current version is not yet suitable for publication in Sensors journal for the following main reasons:
- The novelty is insufficient. Although the authors propose a viewpoint planning strategy based on feature cluster constrained spaces, its basic idea is similar to existing methods based on synthesized viewpoint spaces in the literature, without showing enough innovative points.
- The method lacks generality. The authors' method relies on prior modeling of the features, which limits its applicability in real applications. Modeling of unknown features in unstructured environments remains a challenge.
- The experimental evaluation is not systematic enough. This paper conducts experiments on two different machine vision systems, but the test cases are still limited. The authors should evaluate the method on more diverse types of visual tasks to demonstrate its generalizability.
- The paper organization needs improvement. The logical connection between some sections is not clear enough. Some mathematical formulas also lack necessary explanations for readers to understand.
- The language description needs refinement. Some sentences in the paper are lengthy and not concise enough. I suggest the authors further improve the language and organization during revision to make the paper more fluent and easier to read.
In summary, this paper still has some deficiencies in terms of novelty, applicability, evaluation, and organization. I suggest the authors make further improvements in these aspects before considering resubmission.
See above
Author Response
See the attached document.
